# Sparse Gaussian Process Hyperparameters: Optimize or Integrate?

**Vidhi Lalchand**
Department of Physics
University of Cambridge
`vr308@cam.ac.uk`

**Wessel P. Bruinsma**
Microsoft Research AI4Science
`wbruinsma@microsoft.com`

**David R. Burt**
LIDS
Massachusetts Institute of Technology
`dburt@mit.edu`

**Carl E. Rasmussen**
Department of Engineering
University of Cambridge
`cer54@cam.ac.uk`

## Abstract

The kernel function and its hyperparameters are the central model selection choice in a Gaussian process [Rasmussen and Williams, 2006]. Typically, the hyper-parameters of the kernel are chosen by maximising the marginal likelihood, an approach known as *Type-II maximum likelihood* (ML-II). However, ML-II does not account for hyperparameter uncertainty, and it is well-known that this can lead to severely biased estimates and an underestimation of predictive uncertainty. While there are several works which employ a fully Bayesian characterisation of GPs, relatively few propose such approaches for the sparse GPs paradigm. In this work we propose an algorithm for sparse Gaussian process regression which leverages MCMC to sample from the hyperparameter posterior within the varia-tional inducing point framework of [Titsias, 2009]. This work is closely related to Hensman et al. [2015b], but side-steps the need to sample the inducing points, thereby significantly improving sampling efficiency in the Gaussian likelihood case. We compare this scheme against natural baselines in literature along with stochastic variational GPs (SVGPs) along with an extensive computational analysis.

## 1   Introduction

Gaussian processes (GPs) are a prominent class of models for supervised learning which can quantify uncertainty and incorporate inductive biases in function space via the kernel function. Hand-crafting a kernel function is a powerful way to incorporate prior knowledge. In many instances not all properties of a kernel function can be specified from prior knowledge alone, and parameters are chosen via ML-II. However, defining a complex kernel function with a large number of hyperparameters can make the marginal likelihood prone to multiple local optima and overfitting. Further, several local optima may correspond to priors that do not sensibly model the data. Weakly identified hyperparameters can manifest in flat ridges in the marginal likelihood surface[1] making gradient based optimisation extremely sensitive to starting values [Warnes and Ripley, 1987]. Overall, the ML-II point estimates for the hyperparameters are subject to high variability and underestimate prediction uncertainty.

The problem of ridges in the marginal likelihood surface also does not necessarily go away as more observations are collected. For example, if $f_1$ and $f_2$ are Brownian motions, $\sigma f_1(x/\ell)$ is equal in distribution to $\sqrt{\alpha}\sigma f_2(x/\alpha\ell)$, which means observations do not provide any information about the product $\sigma\ell$. More generally, for a greater class of kernels, including the Matérn–1/2 kernel, $\sigma f_1(x/\ell)$

---

[1]where different combinations of hyperparameters give very similar marginal likelihood values

36th Conference on Neural Information Processing Systems (NeurIPS 2022).

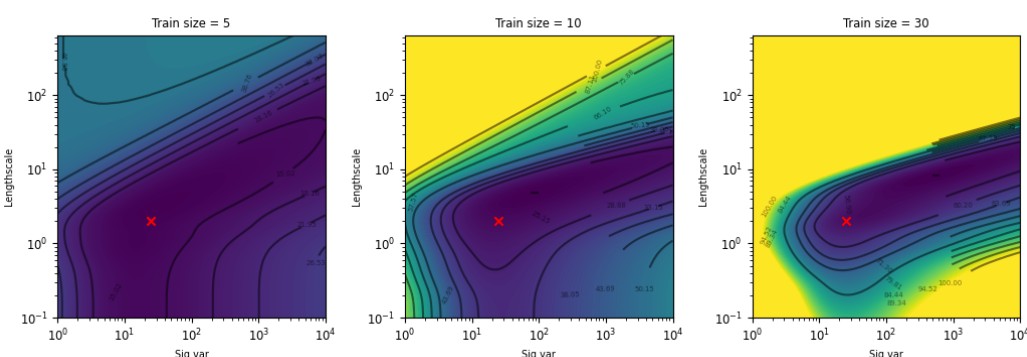

Figure 1: Negative log marginal likelihood surface as a function of two hyperparameters: $\sigma_f^2$ and $l$ for a squared exponential kernel and 1d function. The red cross indicates the true hyperparameters. The hyperparameters selected via gradien- based optimisation are sensitive to the initialisation due to the long ridge of almost identical height at values of the hyperparameters not concordant with the ground truth.

is *equivalent* to $\sqrt{\alpha}\sigma f_2(x/\alpha\ell)$, which means that is not possible to consistently estimate the product $\sigma\ell$ from data, no matter how many observations are collected in a fixed domain [Chapter 6, Stein, 1999]. This implies that one cannot estimate the individual hyperparameters $(\sigma, \ell)$ consistently. It also motivates why there can be benefits to estimating the hyperparameter posterior even in large data regimes, and ML-II may be insufficient. A more satisfactory treatment of hyperparameters involves placing a prior over the hyperparameters and performing Bayesian inference to compute a (hyper)posterior. For large datasets, this motivates using scalable GP inference (e.g. sparse methods) in conjunction with Markov chain Monte Carlo (MCMC) for the hyperparametrs.

The Bayesian treatment of weakly identified hyperparameters may also be fraught with difficulties. Gradient-based samplers like Hamiltonian Monte Carlo [Neal et al., 2011] and its variants have difficulty navigating regions of high curvature and flat ridges where the gradient offers no information for transition [Betancourt, 2017]. This leads to over-concentration of samples from the flat region. (Usually, this can at least partially be rectified with informative priors.) These pathologies are also typical of other hierarchical models [Betancourt and Girolami, 2015]. Figure 1 shows that the GP marginal likelihood surface can manifest these pathologies. The evidence lower bound (ELBO) used in hyperparameter selection for variational sparse GPs relying on inducing points inherits similar, or even less favourable, characteristics to the exact marginal likelihood. As a result, for weakly informative priors, gradient based samplers are susceptible to getting *stuck* at the boundary of these pathological regions hence biasing the sample estimates. The effective sample size metric, used for diagnosing mixing in MCMC, is indicative of this behaviour when directly observing the phase space of the target distribution, but is infeasible in high-dimensions.

Historical justification for ML-II (also called the *evidence* framework) comes from [MacKay, 1994] which highlighted several conditions for ML-II to yield reasonable estimates for the hyperparameters. Crucially, the evidence is unlikely to manifest multiple local optima for a model well-matched to the data and with a high signal-to-noise ratio. Transferring this insight to the Gaussian process paradigm we show how the evidence can have a significant tail or no well-defined maximum in settings with a low signal-to-noise ratio which arises with high aleatoric uncertainty or sparse data, frequently both. In these settings, a point estimate may not adequately summarise the hyperparameter posterior and the benefits of marginalisation stand out. The situation is only exacerbated in high-dimensions (i.e. when there are many hyperparameters) where increasingly more volume of the posterior is captured in a thin shell making the density peak extremely unrepresentative of the posterior. Unlike the likelihood of parameters, the marginal likelihood inherently contains a trade-off between the data-fit and complexity penalty term. This is one of the main properties that makes the marginal likelihood objective a viable choice for model selection [Rasmussen and Williams, 2006]. For example, for the Gaussian process regression model,

$$y_n = f(x_n) + \epsilon_n,\ \epsilon_n \sim \mathcal{N}(0, \sigma^2),\ f \sim \mathcal{GP}(0, k_\theta) \tag{1}$$

the marginal likelihood takes the form,

$$\log p(\boldsymbol{y}|\boldsymbol{\theta}) = \log \int p(\boldsymbol{y}|f)p(f|\boldsymbol{\theta})df = c \overbrace{-\tfrac{1}{2}\boldsymbol{y}^T(K_\theta + \sigma^2 I)^{-1}\boldsymbol{y}}^{\text{data fit term}} \overbrace{-\tfrac{1}{2}|K_\theta + \sigma^2 I|}^{\text{complexity penalty}} \tag{2}$$

where $c$ is a constant, $p(\boldsymbol{y}|f)$ denotes the data likelihood and $\boldsymbol{\theta}$ denotes kernel hyperparameters. This trade-off is a well-established idea which embodies the *automatic Occam's razor* effect [Rasmussen and Ghahramani, 2001] where models well-suited to the data are automatically selected just by using the marginal likelihood objective.

This may seem to contradict earlier claims regarding overfitting, but by shying away from dealing with the hyperparameter posterior we risk overfitting even with the marginal likelihood approach. In other words, the evidence framework subdues the overfitting effect induced by the canonical maximum likelihood approach which does not have any complexity penalty term. The parameters in the canonical approach are free to fit the data as well as possible making them prone to overfitting and poor generalisation. Overparameterized kernels based on neural networks like deep kernel models [Wilson et al., 2016] are well known to exacerbate overfitting [Ober et al., 2021].

The main motivation for this work is to highlight that fully Bayesian schemes in sparse Gaussian process models are practically beneficial. While several works in the literature employ fully Bayesian scheme of integrating out the hyperparameters (see table 1), in this work we attempt to analyse them in an orthogonal direction, focusing on comparison with the evidence framework and other benchmarks by extending the main sparse variational formulation in the literature Titsias [2009]. We present a generalised inference scheme for fully Bayesian GP regression and counteract some of the computational cost of sampling both inducing variables and hyperparameters by deriving a *doubly collapsed* bound which selects the optimal distribution over the inducing points analytically and targets the kernel hyperparameters with HMC.

## 2 Related Work

Table 1: Existing literature on fully Bayesian inference in GPs, sparse GPs and generic likelihoods.

| Index | Reference | Sparse | Posterior ($\boldsymbol{f}/\boldsymbol{u}$) | Posterior ($\boldsymbol{\theta}$) | Methods |
|---|---|---|---|---|---|
| 1. | Murray and Adams [2010a] | ✗ | sampling / NA | sampling | Slice Sampling |
| 2. | Filippone et al. [2013] | ✗ | sampling / NA | sampling | MH + HMC + MA-LA |
| 3. | Filippone and Girolami [2014] | ✗ | Gaussian / NA | sampling | Deterministic + Pseudo-Marginal |
| 4. | Hensman et al. [2015b] | ✓ | Gaussian / sampling | sampling | HMC |
| 5. | Bui et al. [2018] | ✓ | Gaussian / (sampling & VFE) | sampling / VFE | MCMC |
| 6. | Lalchand and Rasmussen [2020] | ✗ | Gaussian / NA | sampling / VFE | NUTS / VI |
| 7. | Rossi et al. [2021] | ✓ | Gaussian / sampling | sampling | SG-HMC |
| 8. | Simpson et al. [2021] | ✗ | Gaussian / NA | sampling | NUTS / Nested Sampling |
| 9. | This work | ✓ | Gaussian / (sampling & VFE) | sampling | HMC / NUTS |

Fully Bayesian Gaussian processes have been used by several authors spawning several variants. In early accounts, Neal [1998], Williams and Rasmussen [1996] explore the integration over covariance hyperparameters using HMC in the regression setting. Barber and Williams [1997] extend this to the classification setting using HMC for sampling in the hyperparameter space and Laplace approximation for the integrand over function values. Murray and Adams [2010a] and Filippone et al. [2013] focused on MCMC schemes to sample covariance hyperparameters in conjunction with latent function values, mainly mitigating the coupling effect through reparameterisation. Hensman et al. [2015b] considered joint sampling of inducing variables and hyperparameters from the optimal variational posterior distribution while [Bui et al., 2018] consider inference schemes for fully Bayesian sparse GPs in a streaming setting. More recently, Rossi et al. [2021] studied fully Bayesian sparse GPs using SG-HMC. Rossi et al. [2021] modify the generative model by adding a prior over the inducing inputs, and peform inference using SG-HMC over the joint $(Z, \boldsymbol{u}, \theta)$ space. We list the most recent works in Table 1.

## 3 Background

Let $f \sim \mathcal{GP}(0, k_{\boldsymbol{\theta}})$ be a Gaussian process prior with kernel function $k_{\boldsymbol{\theta}}$ depending on hyperparameters $\boldsymbol{\theta}$. We are given noisy observations $\boldsymbol{y} = (y_n)_{n=1}^{N} \subseteq \mathbb{R}$ of $\boldsymbol{f} = (f(\boldsymbol{x}_n))_{n=1}^{N}$ at input data $X = (\boldsymbol{x}_n)_{n=1}^{N} \subseteq \mathbb{R}^D$. We consider a Gaussian likelihood which factorises over the data, $p(\boldsymbol{y}|f) = \prod_{n=1}^{N} \mathcal{N}(y_n|f_n, \sigma^2)$. We wish to compute the posterior $p(f|\boldsymbol{y}, \boldsymbol{\theta})$. In this section, we recapitulate the canonical inducing variable approximation of $p(f|\boldsymbol{y}, \boldsymbol{\theta})$ by Titsias [2009] and its extension to a Bayesian treatment of the hyperparameters.

## 3.1 Sparse variational inference in Gaussian processes

Following Titsias [2009], we consider a set of inducing variables $\boldsymbol{u} = \{f(\boldsymbol{z}_m)\}_{m=1}^M \subseteq \mathbb{R}$ at inducing inputs $Z = \{\boldsymbol{z}_m\}_{m=1}^M, \boldsymbol{z}_m \in \mathbb{R}^d$. The complete generative model can then be factored as,

$$p(\boldsymbol{y}, f, \boldsymbol{u}|\boldsymbol{\theta}) = p(\boldsymbol{y}|f, \boldsymbol{\theta})p(f|\boldsymbol{u}, \boldsymbol{\theta})p(\boldsymbol{u}|\boldsymbol{\theta}) \tag{3}$$

We approximate the posterior $p(f, \boldsymbol{u}|\boldsymbol{y}, \boldsymbol{\theta})$ with a variational distribution:

$$p(f, \boldsymbol{u}|\boldsymbol{y}, \boldsymbol{\theta}) \approx q(f, \boldsymbol{u}|\boldsymbol{\theta}) = p(f|\boldsymbol{u}, \boldsymbol{\theta})q(\boldsymbol{u}) \tag{4}$$

where $q(\boldsymbol{u})$ is chosen to minimise the Kullback–Leibler divergence $\text{KL}(q(f|\boldsymbol{\theta}) \| p(f|\boldsymbol{y}, \boldsymbol{\theta}))$. Minimising this KL divergence corresponds to maximising the evidence lower bound [Matthews et al., 2016], henceforth called the ELBO:

$$\text{ELBO}(q(\boldsymbol{u}), \boldsymbol{\theta}) = \mathbb{E}_{q(f|\boldsymbol{\theta})}[\log p(\boldsymbol{y}|f, \boldsymbol{\theta})] - \text{KL}(q(\boldsymbol{u}|\boldsymbol{\theta}) \| p(\boldsymbol{u}|\boldsymbol{\theta})) \tag{5}$$

Because the ELBO still depends on $q(\boldsymbol{u})$, this bound is called *uncollapsed*. Hensman et al. [2013, 2015a] let $q(\boldsymbol{u})$ be a Gaussian, which is optimal if the likelihood is Gaussian [Titsias, 2009], approximate the expectation using Monte Carlo, and maximise the ELBO using stochastic optimisation. On the other hand, if the likelihood is Gaussian, Titsias [2009] computes the optimal form for $q(\boldsymbol{u})$ directly:

$$q^*(\boldsymbol{u}|\boldsymbol{\theta}) = \text{argmax}_{q(\boldsymbol{u})} \, \text{ELBO}(q(\boldsymbol{u}), \boldsymbol{\theta}) \propto p(\boldsymbol{u}|\boldsymbol{\theta}) \exp \mathbb{E}_{p(f|\boldsymbol{u}, \boldsymbol{\theta})}[\log p(\boldsymbol{y}|f, \boldsymbol{\theta})] \tag{6}$$

Plugging $q^*(\boldsymbol{u}|\boldsymbol{\theta})$ back into equation (5), the resulting bound is called *collapsed*, because it now only depends on $\boldsymbol{\theta}$ and $Z$. The collapsed bound, denoted $\mathcal{L}_{\boldsymbol{\theta}, Z}$, is the objective that Titsias [2009] proposes:

$$\log p(\boldsymbol{y}|\boldsymbol{\theta}) \geq \log \mathcal{N}(\boldsymbol{y}; \boldsymbol{0}, K_{nm}K_{mm}^{-1}K_{mn} + \sigma^2 I) - \frac{1}{2\sigma^2}\text{Tr}(K_{nn} - K_{nm}K_{mm}^{-1}K_{mn}) =: \mathcal{L}_{\boldsymbol{\theta}, Z}, \tag{7}$$

where $K_{nn}$ is the prior covariance matrix of $\mathbf{f}$, $K_{mm}$ is the prior covariance matrix over $\mathbf{u}$ and $K_{nm}$ is crosss-covariance matrix formed by $\mathbf{f}$ and $\mathbf{u}$. Using the collapsed bound, approximate ML-II consists of finding,

$$\boldsymbol{\theta}^* \in \text{argmax}_{\boldsymbol{\theta}, Z} \, \mathcal{L}_{\boldsymbol{\theta}, Z}. \tag{8}$$

Predictions at new functions values can be made in $O(M^2)$ after an initial cost of $O(NM^2)$. Under certain assumptions on the data generating process, even when $M \ll N$, the approximate posterior closely resembles the posterior, and equation (7) is a provably accurate approximation to equation (2) [Burt et al., 2020].

## 3.2 Bayesian treatment of hyperparameters and sparse methods

The extension of the sparse variational framework to a Bayesian treatment of the hyperparameters has been previously considered by Hensman et al. [2015b]. Extend the generative model with a prior $p(\boldsymbol{\theta})$ over the hyperparameters $\boldsymbol{\theta}$, and let the variational approximation of the posterior $p(f, \boldsymbol{u}, \boldsymbol{\theta}|\boldsymbol{y})$ be $q(f, \boldsymbol{u}, \boldsymbol{\theta}) = p(f|\boldsymbol{u}, \boldsymbol{\theta})q(\boldsymbol{u}, \boldsymbol{\theta})$. The analogue of equation (5) is

$$\text{ELBO}(q(\boldsymbol{u}, \boldsymbol{\theta})) = \mathbb{E}_{q(f, \boldsymbol{\theta})}[\log p(\boldsymbol{y}|f, \boldsymbol{\theta})] - \mathbb{E}_{q(\boldsymbol{\theta})}[\text{KL}(q(\boldsymbol{u}) \| p(\boldsymbol{u}|\boldsymbol{\theta}))] - \text{KL}(q(\boldsymbol{\theta}) \| p(\boldsymbol{\theta})) \tag{9}$$

and the optimal form for $q(\boldsymbol{u}, \boldsymbol{\theta})$ can again be determined:

$$q^*(\boldsymbol{u}, \boldsymbol{\theta}) \propto p(\boldsymbol{u}, \boldsymbol{\theta}) \exp \mathbb{E}_{p(f|\boldsymbol{u}, \boldsymbol{\theta})}[\log p(\boldsymbol{y}|f, \boldsymbol{\theta})]. \tag{10}$$

The distribution $q^*(\boldsymbol{u}, \boldsymbol{\theta})$ does not have a closed form, and for general likelihoods, Hensman et al. [2015b] propose to approximate the expectation in (10) with quadrature and to sample from $q^*(\boldsymbol{u}, \boldsymbol{\theta})$ using HMC. While this approach is quite general, in the case of Gaussian regression, it vastly increases the dimensionality of the state space over which HMC must be run relative to HMC in GPR, since the $\boldsymbol{u}$ are sampled in addition to the $\boldsymbol{\theta}$. This increases the cost of the procedure, and impacts the success of the sampler.

An alternative approach to approximately inferring hyperparameters is to assume a parametric form for $q(\boldsymbol{u}, \boldsymbol{\theta})$ and maximise equation (9) with respect to the variational parameters. Bui et al. [2018]

Table 2: Comparison of a variety of approaches to approximating the posterior over hyperparameters in Gaussian process regression. Compares the quality of the posterior (QUALITY); the time complexity per iteration (TIME/IT.); the memory complexity per iteration (MEM./IT.); the number of parameters and/or variables (PARS/VARS); and whether the approach supports non-Gaussian likelihoods (LIK.).

| APPROACH | QUALITY | TIME/IT. | MEM./IT. | PARS/VARS | LIK. |
|---|---|---|---|---|---|
| Maximum a posteriori [MacKay, 1994] | − | $n^3$ | $n^2$ | $n_\theta$ | ✗ |
| VI | | | | | |
|   Inducing points; non-collapsed [Titsias and Lázaro-Gredilla, 2014] | ± | $nm^2$ | $m^2$ | $n_\theta^2 + m^2$ | ✓ |
|   Inducing points; collapsed [Bui et al., 2018] | ± | $nm^2$ | $m^2$ | $n_\theta^2$ | ✗ |
| SAMPLING | | | | | |
|   Exact with Gaussian lik.[Simpson et al., 2021] | + | $n^3$ | $n^2$ | $n_\theta$ | ✗ |
|   Exact with non-Gaussian lik.[Murray and Adams, 2010b] | + | $n^3$ | $n^2$ | $n_\theta + n$ | ✓ |
|   Inducing points; non-collapsed [Hensman et al., 2015b] | ± | $m^3$ | $m^2$ | $n_\theta + m$ | ✓ |
|   Inducing points; collapsed (ours) | ± | $nm^2$ | $m^2$ | $n_\theta$ | ✗ |

took such an approach, assuming that $q(\boldsymbol{u}, \boldsymbol{\theta}) = q(\boldsymbol{u})q(\boldsymbol{\theta})$, with both distributions Gaussian. Similar approaches have been applied to variational inference in state-space modelling, sometimes leveraging the optimal form of $q(\boldsymbol{u}|\boldsymbol{\theta})$ discussed earlier.

The QUALITY column in Table 2 indicates the ability of the method to faithfully represent the hyperparameter posterior. If VI is run to convergence, a potentially significant amount of error will be incurred by the Gaussian approximation to the non-Gaussian posterior over the hyperparameters (red). On the opposite extreme, if no sparsity assumption is made, MCMC over the hyperparameters without sparse approximations is asymptotically consistent (green). The inducing point approximations combined with MCMC lie somewhere in-between these methods (yellow).

## 3.3 Making predictions

The predictive posterior distribution for unknown test inputs $X^*$ integrates over the joint posterior,

$$p(\boldsymbol{f}^*|\boldsymbol{y}) \approx \int p(\boldsymbol{f}^*|\boldsymbol{f}, \boldsymbol{u}, \boldsymbol{\theta})p(\boldsymbol{f}|\boldsymbol{u}, \boldsymbol{\theta})q(\boldsymbol{u}|\boldsymbol{\theta})q(\boldsymbol{\theta})d\boldsymbol{f}d\boldsymbol{u}d\boldsymbol{\theta}, \tag{11}$$

where we have suppressed the conditioning over inputs $X, X^*$ for brevity. The inner integral simplifies to $\int p(\boldsymbol{f}^*|\boldsymbol{f}, \boldsymbol{u}, \boldsymbol{\theta})p(\boldsymbol{f}|\boldsymbol{u}, \boldsymbol{\theta})d\boldsymbol{f} = p(\boldsymbol{f}^*|\boldsymbol{u}, \boldsymbol{\theta})$. We discuss the predictive posterior in such models in section 4.3.

# 4 Fully Bayesian SGPR with HMC: Doubly collapsed formulation

In the previous section, we observed that a major drawback of the approach taken in Hensman et al. [2015b] is the need to sample $\boldsymbol{u}$, which for high-dimensional inputs or in cases where many inducing points are needed could introduce thousands of additional variables to sample. In this section, we leverage the optimal form of $q(\boldsymbol{u}|\boldsymbol{\theta})$ derived in Titsias [2009] to alleviate this sampling problem.

## 4.1 Collapsing the evidence lower bound (again)

We first derive the lower bound for this formulation and provide pseudo-code for the algorithm in Algorithm 1. Following the usual derivation of the ELBO,

$$\log p(\boldsymbol{y}) \geq \int q(\boldsymbol{\theta}) \log p(\boldsymbol{y} \,|\, \boldsymbol{\theta})d\boldsymbol{\theta} - \mathrm{KL}(q(\boldsymbol{\theta}) \,\|\, p(\boldsymbol{\theta})) \tag{12}$$

$$\geq \int q(\boldsymbol{\theta})\mathcal{L}_{\boldsymbol{\theta},Z}d\boldsymbol{\theta} - \mathrm{KL}(q(\boldsymbol{\theta}) \,\|\, p(\boldsymbol{\theta})) = \int q(\boldsymbol{\theta}) \log \frac{M_{\boldsymbol{\theta},Z}p(\boldsymbol{\theta})}{q(\boldsymbol{\theta})}d\boldsymbol{\theta} =: \mathcal{L}_Z^*(q(\boldsymbol{\theta})), \tag{13}$$

where $\log p(\boldsymbol{y} \,|\, \boldsymbol{\theta}) \geq \mathcal{L}_{\boldsymbol{\theta},Z}$ with $\mathcal{L}_{\boldsymbol{\theta},Z}$ defined in equation (7), and where we assign $M_{\boldsymbol{\theta},Z} = e^{\mathcal{L}_{\boldsymbol{\theta},Z}}$.

### 4.1.1 Deriving $q^*(\boldsymbol{\theta})$

We can interpret $\mathcal{L}_Z^*(q(\boldsymbol{\theta}))$ as a negative KL divergence as long as we account for a normalisation constant $C_Z = \int M_{\boldsymbol{\theta},Z}p(\boldsymbol{\theta})d\boldsymbol{\theta}$ for the un-normalised numerator $M_{\boldsymbol{\theta},Z}p(\boldsymbol{\theta})$. Hence, we can re-write $\mathcal{L}_Z^*(q(\boldsymbol{\theta}))$ as,

$$\mathcal{L}_Z^*(q(\boldsymbol{\theta})) = \log C_Z - \mathrm{KL}(q(\boldsymbol{\theta}) \,\|\, q^*(\boldsymbol{\theta})) \tag{14}$$

where $q^*(\boldsymbol{\theta}) = M_{\boldsymbol{\theta},Z} p(\boldsymbol{\theta})/C_Z$. By inspecting equation (14), we observe that the optimal variational distribution over $\boldsymbol{\theta}$ is given by $q^*(\boldsymbol{\theta})$.[2] Crucially, by sampling directly from $q^*(\boldsymbol{\theta})$ using MCMC we eliminate the need to sample the variables $\boldsymbol{u}$. By evaluating $\mathcal{L}_Z^*$ at $q^*(\boldsymbol{\theta})$, we find the *doubly collapsed* ELBO $\mathcal{L}_Z^{**} := \mathcal{L}_Z^*(q^*(\boldsymbol{\theta})) = \log C_Z$. Although the value of $\mathcal{L}_Z^{**}$ is computationally intractable, given samples $(\boldsymbol{\theta}_j)_{j=1}^J$ from $q^*(\boldsymbol{\theta})$, gradients of $\mathcal{L}_Z^{**}$ with respect to $Z$ can be estimated using the stochastic estimate of the canonical ELBO equation (7): using the chain rule,

$$\frac{\mathrm{d}}{\mathrm{d}Z}\mathcal{L}_Z^{**} = \frac{\partial}{\partial Z}\mathcal{L}_Z^*(q)\bigg|_{q=q^*(\boldsymbol{\theta})} + \left\langle \frac{\delta}{\delta q}\mathcal{L}_Z^*(q)\bigg|_{q=q^*(\boldsymbol{\theta})}, \frac{\partial}{\partial Z}q^*(\boldsymbol{\theta}) \right\rangle \approx \frac{1}{J}\sum_{j=1}^J \frac{\partial}{\partial Z}\mathcal{L}_{\boldsymbol{\theta}_j,Z} \qquad (15)$$

where $\frac{\delta}{\delta q}\mathcal{L}_Z^*(q)$ is the functional derivative of $\mathcal{L}_Z^*(q)$ with respect to $q$, which is zero at $q = q^*(\boldsymbol{\theta})$, because $q^*$ optimises $\mathcal{L}_Z^*$ (it is a critical point, so the derivative is zero). Further, the partial derivative of $\mathcal{L}_Z^*$ with respect to $Z$ concerns just the first term of the LHS of equation (13) as the KL term $\mathrm{KL}(q(\boldsymbol{\theta}) \,\|\, p(\boldsymbol{\theta}))$ is independent of $Z$.

## 4.2 Performing approximate inference

We deploy HMC to (approximately) sample from the optimal variational posterior $q^*(\boldsymbol{\theta})$ along with optimising the inducing inputs $Z$ in a hybrid scheme. We alternate between the two steps allocating longer intervals for optimising $Z$ for every HMC sampling run for the hyperparameters. We note that this hybrid scheme is much more computationally efficient than sampling $\boldsymbol{u}$ and $\boldsymbol{\theta}$ jointly where one has to tackle the coupling between inducing variables and hyperparameters in joint space. Further, joint sampling is only feasible for moderate number of inducing variables while this scheme can scale to much larger datasets as the efficiency of sampling in the hyperparameter space is only dependent on the dimensionality of the hyperparameter space rather than the number of inducing variables. The entire inference scheme is summarized in Algorithm 1. The warm-start strategy (of optimizing both $(Z, \boldsymbol{\theta})$ jointly for a few gradient steps) is used to find a good region for the sampler to initialise $\boldsymbol{\theta}$.

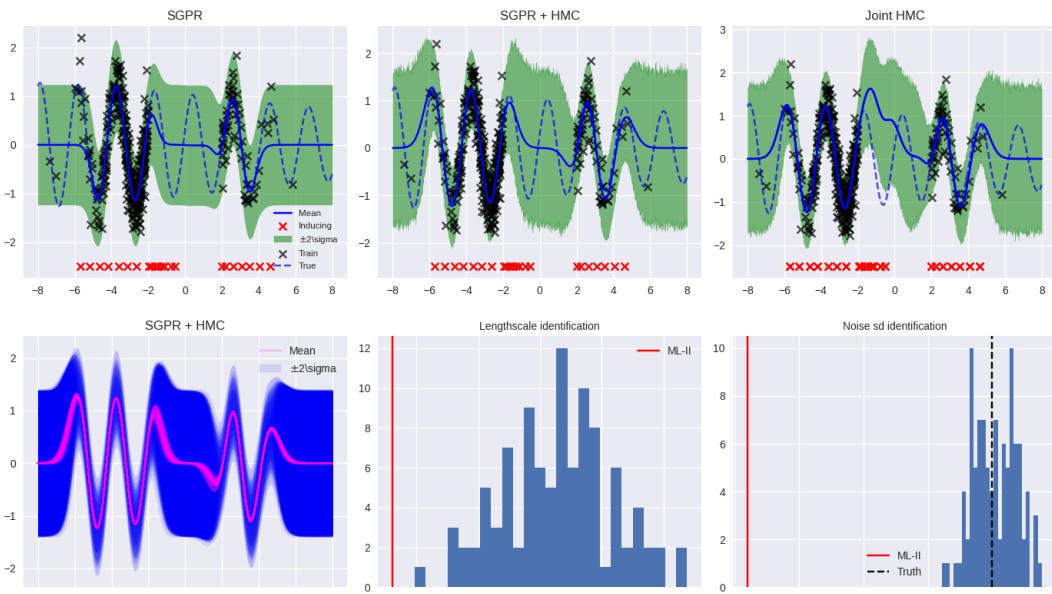

Figure 2: Top - 1d regression with Left: SGPR, Middle: SGPR + HMC, Right: Joint HMC [Hensman et al., 2015b], Below - Left: Samples from the mixture posterior, Middle: Length-scale distribution under SGPR+HMC and ML-II. Note that the data is generated through a parameteric function and hence there is no ground truth lengthscale. Right: Noise std. deviation distribution from SGPR+HMC and ML-II.

---

[2]KL divergence reaches its minimal value of zero when the two input probability distributions are equal, and we seek to maximise $\mathcal{L}_Z^*(q(\boldsymbol{\theta}))$ which entails minimizing the KL.

---
**Algorithm 1** Fully Bayesian Sparse GPR with HMC
---
1: **Input:** ELBO objective $\mathcal{L}_{\theta,Z} := \mathcal{L}(\theta, Z)$ (equation (7)), gradient based optimiser `optim()`
2:
3: **procedure** WARM-START
4:     **for** fixed number of steps **do**
5:         Gradient step: $Z, \theta \longleftarrow \texttt{optim}(\mathcal{L}(\theta, Z))$
6: **return** initial values $Z_{init}$ and $\theta_{init}$
7:
8: **procedure** TRAIN
9:     `## Initialisation protocol`
10:     ▶ Initialise $\mathcal{L}_{\theta,Z}$ at warm-start values $\mathcal{L}(\theta_{init}, Z_{init})$, lets call this $\hat{\mathcal{L}}$
11:     ▶ Freeze kernel hyperparameters in the ELBO objective by setting `requires_grad=False`.
12:
13:     **while** not converged **do**
14:         **for** $t = 1 \ldots T$ **do**  `## start of training loop`
15:
16:             • Gradient step: $Z_{opt} \longleftarrow \texttt{optim}(\hat{\mathcal{L}})$ (`## equation (15) shows the validity of taking the derivative of the stochastic ELBO`)
17:             **if** $t \bmod L == 0$ **then**
18:                 (`## For every L gradient steps`)
19:
20:                 • Draw $J$ samples from the optimal hyperparameter variational distribution
21:                 $\log q^*(\theta) \propto \mathcal{L}(\theta, Z_{opt}) + \log p(\theta)$ keeping $Z_{opt}$ fixed.

$$(\theta_j, p_j) \overset{\text{HMC}}{\longleftarrow} \mathcal{H}(\theta, p), \text{(where } \mathcal{H} \text{ is the Hamiltonian)}$$

22:                 where $p$ denotes the zero-mean auxilliary momentum variable in phase space with
23:                 the same dimensionality as $\theta$.
24:                 • Compute stochastic ELBO $\hat{\mathcal{L}} = \frac{1}{J}\sum_{j=1}^{J} \mathcal{L}(\theta_j, Z_{opt})$, where $\theta_j \sim q^*(\theta)$
25: **return** $Z_{opt}, \{\theta\}_{j=1}^{J}$
---

## 4.3 Predictive posterior

It is ultimately the posterior predictive (PP) distribution that is of interest rather than point estimates or the hyperparameter posterior. The Bayesian sparse GP predictive posterior entails integrating out the posterior over inducing variables $u$ and hyperparameters $\theta$. Once we have performed inference, we can approximate this directly,

$$p(\boldsymbol{f}^*|\boldsymbol{y}) = \int \int p(\boldsymbol{f}^*|\boldsymbol{u}, \theta)p(\boldsymbol{u}, \theta|\boldsymbol{y})d\boldsymbol{u}d\theta = \int \int p(\boldsymbol{f}^*|\boldsymbol{u}, \theta)p(\boldsymbol{u}|\theta, \boldsymbol{y})p(\theta|\boldsymbol{y})d\boldsymbol{u}d\theta \tag{16}$$

$$\approx \int \int p(\boldsymbol{f}^*|\boldsymbol{u}, \theta)q^*(\boldsymbol{u}|\theta, \boldsymbol{y})p(\theta|\boldsymbol{y})d\boldsymbol{u}d\theta = \int \mathcal{N}(\boldsymbol{f}^*|A\boldsymbol{m}^*, K_{**} + A(S^* - K_{mm})A^T)p(\theta|\boldsymbol{y})d\theta$$

where $q^*(\boldsymbol{u}|\theta, \boldsymbol{y}) = \mathcal{N}(\boldsymbol{m}^*, S^*)$ is the optimal Gaussian variational distribution and $A = K_{*m}K_{mm}^{-1}$ under the SGPR scheme (Section 4) is available in closed form with $\boldsymbol{m}^* = \sigma^{-2}(K_{mm} + \sigma^{-2}K_{mn}K_{nm})$ and $S^* = K_{mm}(K_{mm} + \sigma^{-2}K_{mn}K_{nm})^{-1}K_{mm}$. In either case, the internal integral with respect to the inducing variables is analytic and the outer integral can be estimated using the samples collected from HMC to perform Monte Carlo estimation. This yields a mixture of Gaussians,

$$p(\boldsymbol{f}^*|\boldsymbol{y}) \approx \frac{1}{J}\sum_{j=1}^{J}\mathcal{N}(\boldsymbol{\mu}^{\theta_j}, \Sigma^{\theta_j}), \qquad \theta_j \sim q^*(\theta), \tag{17}$$

$$\boldsymbol{\mu}^{\theta_j} = A^{(\theta_j)}\boldsymbol{m}^{(\theta_j)}, \qquad \Sigma^{\theta_j} = K_{**}^{\theta_j} + A^{(\theta_j)}(S^{\theta_j} - K_{mm}^{(\theta_j)})A^{T^{(\theta_j)}} \tag{18}$$

where $J$ samples are (approximately) drawn from $q^*(\theta)$ via HMC. The distribution inside the summation is the Gaussian posterior predictive distribution for fixed hyperparameters with identical mixing proportions. The compute cost for the predictive posterior scales the sparse GPR cost linearly in the number of samples. The Monte Carlo approximation costs $\mathcal{O}(JNM^2)$ for $M$ inducing points and $J$ hyperparameter samples.

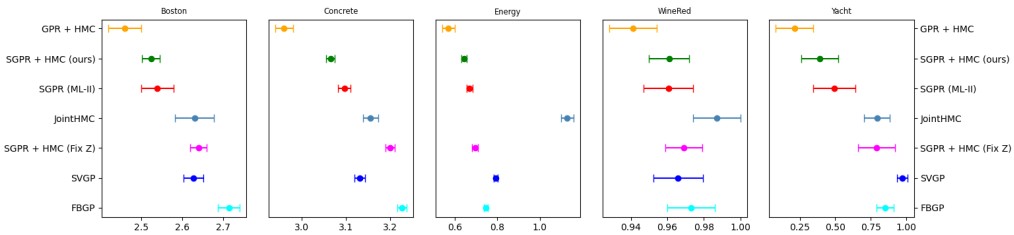

Figure 3: Negative test log likelihoods with standard error of mean across 10 splits with 80% of the data reserved for training. Our method is SGPR + HMC.

## 5 Experiments

In the previous section, we discussed a hybrid scheme which leverages MCMC within the variational sparse inducing variable formulation leading to fully Bayesian sparse Gaussian processes. In the experiments we demonstrate the feasibility of this scheme relative to several benchmarks and assess regression performance on a 1-dimensional illustrative example and a range of other datasets. We also compare with exact GPs where the cost of using a gradient based sampler is prohibitive for even moderately large datasets requiring several inversions of the full covariance matrix. We show that using the doubly collapsed scheme proposed in this work is a much more attractive alternative for large datasets as compared to direct HMC in GPR, and sampling is more efficient than in the uncollapsed bound used in Hensman et al. [2015b].

We henceforth refer to benchmark methods as follows: **SGPR + HMC** refers to Bayesian GPs with doubly collapsed variational inference with NUTS, as described in section 3 (compatible with Gaussian likelihoods); we benchmark this model against sparse GPs (**SGPR**) [Titsias, 2009] and Stochastic Variational GPs (**SVGP**) both using approximate ML-II [Hensman et al., 2015a]. Additionally, we consider the sparse, joint sampling inference scheme proposed in [Hensman et al., 2015b] which gives a natural benchmark. We call this model **JointHMC**. The **FBGP** method extends the Bayesian treatment to the inducing locations similar to Rossi et al. [2021]; we use NUTS to sample from the joint posterior over $(Z, \boldsymbol{\theta})$. It is not a direct comparison to Rossi et al. [2021] as the latter explores free-form sampling of $\boldsymbol{u}$ along with $(Z, \boldsymbol{\theta})$ while we work with the collapsed bound incorporating the optimal Gaussian variational distribution $q^*(\boldsymbol{u})$.

We also present analysis where we fix inducing point locations at a random subset of the training data (as opposed to interleaving as per Algorithm 1) and only learn hyperparameters using NUTS. We provide several details about the experimental set-up in the supplementary.

### 5.1 One dimensional synthetic data

We sample noisy observations from $f(x) = \sin(3x) + 0.3\cos(\pi x)$ with the constraint $(x < -2)$ and $(x > 2)$. Figure 2 shows the results for SGPR along with the fully Bayesian schemes. We keep data split and noise identical across the three models to facilitate a comparison. While there is significant data to identify the hyperparameters we notice that the models mainly differ in their extrapolation abilities away from the training data. SGPR with ML-II overfits to the training data and recovers a low lengthscale,

Table 3: Prediction performance in 1D synthetic regression across SGPR, SGPR + HMC and JointHMC methods with identical number of inducing points and train/test split.

| Method | SGPR | SGPR + HMC | JointHMC |
|--------|------|-----------|----------|
| RMSE | 0.580 | **0.537** | 0.682 |
| NLPD | 0.214 | **0.065** | 0.74 |

low noise solution while the SGPR + HMC scheme recovers a more moderate fit and performs significantly better in terms of RMSE and NLPD on unseen data Table 3. We note that the JointHMC scheme which samples both $(\boldsymbol{u}, \boldsymbol{\theta})$ overfits in the central missing data region. We use $M = 25$ inducing locations across all methods which are optimised according to the protocol of each method and recover a very similar spatial distribution.

Table 4: A comparison of Sparse GP approaches for UCI benchmarks. RMSE ($\pm$ standard error of mean) evaluated on average of 10 splits with 80% of the data used for training. $\delta$ indicates that the posterior over hyperparameters is approximated by a point estimate under the respective scheme.

| Dataset | $N$ | $d$ | GPR + HMC | SGPR | SGPR + HMC | SVGP | JointHMC | FBGP |
|---|---|---|---|---|---|---|---|---|
| $|M|$ | - | - | - | 100 | 100 | 100 | 100 | 100 |
| $q(\boldsymbol{\theta})$ | - | - | free-form | $\delta$ | free-form (ours) | $\delta$ | free-form | free-form $(Z, \theta)$ |
| Boston | 506 | 13 | 3.049 (0.14) | 3.291 (0.11) | 3.286 (0.09) | 3.619 (0.11) | 3.28 (0.11) | 3.845 (0.103) |
| Concrete | 1030 | 8 | 4.864 (0.12) | 5.459 (0.09) | 5.402 (0.05) | 5.617 (0.09) | 5.612 (0.09) | 6.084 (0.11) |
| Energy | 768 | 8 | 0.441 (0.01) | 0.477 (0.008) | 0.469 (0.009) | 0.500 (0.01) | 0.755 (0.02) | 0.490 (0.011) |
| WineRed | 1599 | 11 | 0.640 (0.01) | 0.636 (0.008) | 0.635 (0.008) | 0.641 (0.007) | 0.641 (0.007) | 0.642 (0.007) |
| Yacht | 308 | 6 | 0.353 (0.03) | 0.412 (0.03) | 0.387 (0.03) | 0.606 (0.04) | 0.794 (0.07) | 0.569 (0.037) |

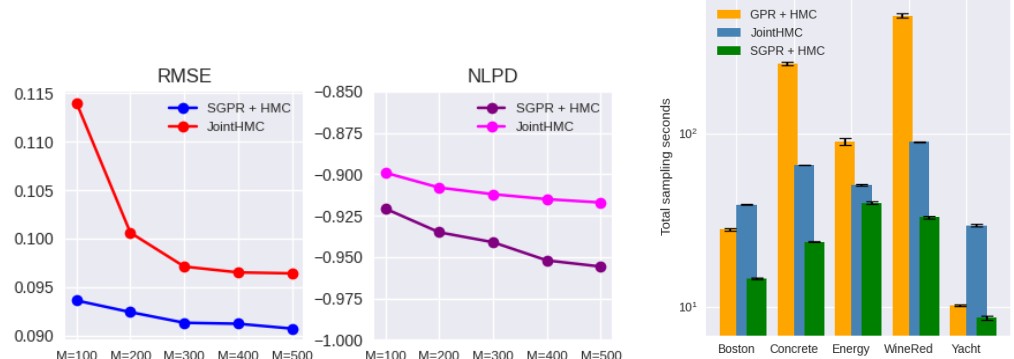

Figure 4: Left: Test RMSE and NLPD for a range of inducing points for the Elevator dataset. Right: Sampling performance measured in terms of the time it took to draw the combined set of samples during the training phase (excluding tuning) as defined by the python standard library *time.perf_counter (wall time)*. We use the `pymc3 pm.NUTS` sampler for GPR + HMC and SGPR + HMC, and `tfp.mcmc.HamiltonianMonteCarlo` for JointHMC [Matthews et al., 2017]. All experiments were conducted on an Intel Core i7-10700 CPU @ 2.90GHz x 16.

## 5.2 UCI regression benchmarks

We compare our approach across methods on 5 standard small to medium-sized UCI benchmark datasets. Following common practice, we use a 20% randomly selected held out test-set [Rossi et al., 2021, Havasi et al., 2018] and scale the inputs and outputs to zero mean and unit standard deviation within the training set (we restore the output scaling for evaluation) [Salimbeni and Deisenroth, 2017]. While we could use any kernel, we choose the RBF-ARD kernel with a lengthscale for each dimension. For consistency we initialise all the inducing locations ($Z$) identically across the methods, i.e. by using the same random subset of training data split. We note that adapting the inducing locations brings serious gains in prediction performance versus keeping them fixed (Figure 3). Further, the JointHMC scheme underperforms SGPR (with ML-II) and SGPR + HMC. This is not surprising given that the JointHMC bound equation (10) does not incorporate the optimal setting for $q(\boldsymbol{u})$ and was originally motivated by the need for a fully Bayesian scheme for generalised likelihoods. The method SGPR + HMC significantly improves upon JointHMC in the specific Gaussian likelihood case.

**Sensitivity to** $M$: We benchmark SGPR+HMC and JointHMC on the Elevator dataset ($N = 16599, D = 18$) which demands a larger $M$. SGPR+HMC outperforms JointHMC for this dataset across different $M$ but the advantage is more pronounced at smaller $M$. Further, our method took 1248 vs. 2109 wall clock sec. for the joint scheme for the same number of hyperparameter samples and 500 inducing points.

## 5.3 Ablation study

In order to understand the efficacy of Algorithm 1 we conduct an ablation study where we perform inference in the same manner, but keeping inducing locations fixed. Algorithmically, this implies that we don't need to compute the stochastic ELBO $\hat{\mathcal{L}}$ and just conduct a single sampling run for the hyperparameters. The results across 10 splits are summarised in Table 5.

Table 5: An ablation study for the doubly collapsed Sparse GPR scheme comparing performance with and without adapting the inducing locations during training. We report test NLPDs and RMSEs over 10 splits.

| Dataset | Metric | Boston | Concrete | Energy | WineRed | Yacht |
|---|---|---|---|---|---|---|
| Fixed Z | RMSE | 3.624 (0.110) | 6.021 (0.12) | 0.499 (0.014) | 0.640 (0.007) | 0.533 (0.036) |
| Adapt Z (ours) | | **3.286** (0.090) | **5.405** (0.07) | **0.469** (0.009) | 0.635 (0.008) | **0.387** (0.030) |
| Fixed Z | NLPD | 2.640 (0.020) | 3.200 (0.06) | 0.696 (0.014) | 0.969 (0.012) | 0.791 (0.130) |
| Adapt Z (ours) | | **2.524** (0.022) | **3.065** (0.01) | **0.644** (0.013) | 0.961 (0.011) | **0.391** (0.146) |

## 5.4 Runtimes

While it is possible to train exact GPR with HMC for datasets of this size (in terms of $N$) it is important to look at the trade-off in terms of compute cost. In Figure 4 we record the average number of wall clock seconds to draw 500 samples under each method. The cost of sampling is $\mathcal{O}(N)$ for SGPR + HMC but $\mathcal{O}(N^3)$ for Exact GPR + HMC. JointHMC deals with a higher dimensional phase space $(\boldsymbol{u}, \boldsymbol{\theta})$ hence requires more tuning. We don't include tuning time for a fair comparison. For further context we report the total training run-time for our scheme alongside ML-II, GPR + HMC and FBGP in Table 6. The hybrid scheme we propose is significantly cheaper to canonical alternatives with virtually no degradation in predictive performance.

Table 6: Wall clock seconds (this counts all the CPU time, including worker processes in BLAS and OpenMP as defined by the python standard library `time` for a single training split).

| Dataset | Boston | Concrete | Energy | WineRed | Yacht |
|---|---|---|---|---|---|
| SGPR (ML-II) | 22.17 (0.21) | 33.06 (0.07) | 30.36 (0.09) | 39.96 (0.87) | 20.41 (0.22) |
| SGPR + HMC (ours) | 29.47 (0.34) | 53.85 (2.36) | 61.60 (1.47) | 60.65 (0.63) | 24.50 (0.40) |
| GPR + HMC | 78.05 (2.36) | 977.40 (13.82) | 326.18 (15.87) | 1426.25 (39.49) | 31.71 (0.59) |
| FBGP | 72.63 (8.29) | 156.31 (4.30) | 259.81 (11.58) | 175.45 (13.14) | 101.92 (2.27) |

## 6  Discussion

The evidence framework continues to be the pre-dominant method for training Gaussian processes since their inception into modern machine learning [Rasmussen and Williams, 2006]. While the marginal likelihood is a compelling model selection objective as it offers an inherent trade-off between data-fit and complexity, it is susceptible to overfitting and other pathologies leading to biased inference. This work builds on existing methods that combine sparse Gaussian process regression based on inducing variables with Bayesian hyperparameter inference.

Bayesian hyperparameter inference in GPs is however intractable and one has to consider balancing between objectives of computational cost, prediction accuracy and robustness of uncertainty intervals. While in straightforward conditions the fully Bayesian approach might be counter-productive, most real-world applications of GPs rely on engineering sophisticated hand-crafted kernels involving many hyperparameters where there risk of overfitting is pronounced and further, harder to detect. A more robust solution is to incorporate prediction intervals that reflect these uncertainties in the model choice. Studying full Bayesian inference in more sophisticated GP models like deep [Damianou and Lawrence, 2013], warped [Snelson et al., 2004] and convolutional GPs [Van der Wilk et al., 2017] will offer greater insight to this question and is an imminent direction of future work.

## Acknowledgements

This research was conducted while WPB and DRB were students at the University of Cambridge. During that time, WPB was supported by the Engineering and Physical Research Council (studentship number 10436152). VL acknowledges funding from the Qualcomm Innovation Fellowship (Europe).

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
