# A  Experimental Set-up

For methods SGPR, SGPR + HMC, JointHMC and Ablation experiment we use the Adam [Kingma and Ba, 2014] optimizer with a learning rate set at 0.01 (we didn't extensively tune for learning rates and 0.01 seemed to give a reasonable performance). We do maintain consistency over the data splits and initialisation values for the inducing locations and hyperparameters across all the methods. Further, all the sparse models use $M = 100$ inducing variables to aid in run-time analysis. All the hyperparameters are initialised at the `gpytorch` default of $\log(2)$ and inducing locations at a random subset of the training data split.

**SGPR + HMC:** We place individual priors over the set of hyperparameters $\{\{l_d\}_{d=1}^D, \sigma_f, \sigma_n\}$ shown in the code block below. During the warm-up phase we optimize both the inducing locations and hyperparamters. We use $J = 100$ samples to construct the stochastic ELBO for the first sampling window along with 500 steps of tuning, thereafter just 10 samples are used every 50 gradient steps. At the end of training we again draw $J = 100$ samples. The intermediate sampling windows do not require elaborate tuning as we persist good initial step-size values from the penultimate chains. Despite this we do expend a few tuning steps in each sampling window as it improved the overall performance of the sampler. The inducing locations are kept fixed during sampling and are only optimized through the stochastic ELBO.

**JointHMC:** As recommended by the authors we use a warm-up phase of 100 gradient steps to optimize inducing locations. Subsequent training happens through the HMC sampler which targets the joint variables $(\boldsymbol{v}, \boldsymbol{\theta})$ (where $\boldsymbol{v}$ is a whitened representation of $\boldsymbol{u}$) with a target acceptance rate of 0.8, path length (number of leapfrog steps) to 10 and an initial step-size of 0.01 with an adaptation rate of 0.1. We use `tfd.Gamma(2.0, 1.0)` for each inidividual kernel hyperparamter.

## A.1  Software & Code

The software for all the methods is largely written in `gpytorch` [Gardner et al., 2018]. For sampling we resort to the auto-tuning NUTS sampler in `pymc3` [Salvatier et al., 2016]. The JointHMC model uses the SGPMC class from `gpflow` [Van der Wilk et al., 2020]. The source code for all the models and experiments is attached with the supplementary.

The code-snippet below shows the straight-forward `pymc3` sampling loop which is triggered at pre-specified intervals.

```
with pm.Model() as model_pymc3:

    ls = pm.Gamma("ls", alpha=2, beta=1, shape=(input_dim,))
    sig_f = pm.HalfCauchy("sig_f", beta=1)

    cov = sig_f ** 2 * pm.gp.cov.ExpQuad(input_dim, ls=ls)
    gp = pm.gp.MarginalSparse(cov_func=cov, approx="VFE")
    sig_n = pm.HalfCauchy("sig_n", beta=1)

    # Z_opt is the intermediate inducing points from the optimisation stage
    y_ = gp.marginal_likelihood("y", X=self.train_x.numpy(), Xu=Z_opt, \\
                                        y=self.train_y.numpy(), noise=sig_n)

    if sampler_params is not None:
        step = pm.NUTS(step_scale = sampler_params['step_scale'])
    else:
        step = pm.NUTS()
    trace = pm.sample(n_samples, tune=tune, chains=1, step=step, \\
                                        return_inferencedata=False)
return trace
```

# B  Further Analysis

## B.1  Comparison with Deep GPs and Neural Network Benchmarks

We additionally compare the performance of our algorithm to 2, 3 and 4 layer deep GPs (DGP 2–4), each with 100 inducing points and point estimation for the hyperparameters [Damianou and Lawrence, 2013], [Salimbeni

and Deisenroth, 2017]. We also compare to a two-layer Bayesian neural network with ReLu activations, 50 hidden units, with inference by probabilistic backpropagation (PBP). The results were taken from Salimbeni and Deisenroth [2017] and Hernández-Lobato and Adams [2015] respectively and follow a very similar data processing scheme for the datasets. We learn the inducing locations $Z$ through optimisation but keep the number of inducing points fixed across all methods.

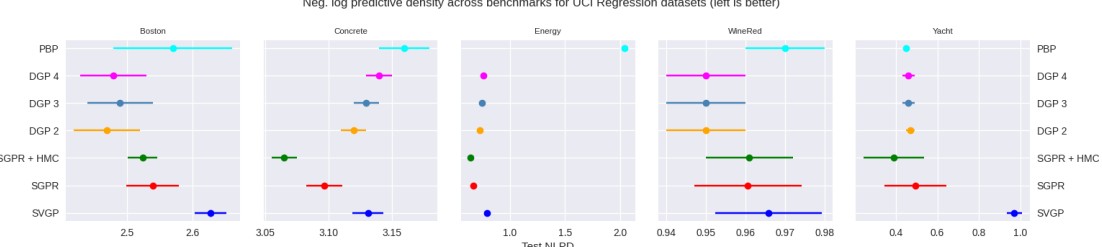

Neg. log predictive density across benchmarks for UCI Regression datasets (left is better)

Figure 5: Negative test log likelihoods with standard error of mean with 80% of the data reserved for training. Our method is SGPR + HMC.

The negative test log-likelihood results are shown in 5. The test log-likelihoods outperform the non-Bayesian counterparts and in most cases perform as well if not better than a multi-layer deep GP with a significantly higher computational cost and intractabilities. Further, the variability across splits is much lower for the HMC method versus SGPR.

## B.2   NUTS Sampling Summary

In the tables below we include the summary statistics of the NUTS sampler for split 4 for each dataset for the SGPR + HMC model. The statistics were computed based on the trace of the final sampling window. The columns hdi_3% and hdi_97% calculate the highest posterior density interval based on marginal posteriors. $\mathtt{ess} = \dfrac{MN}{1 + 2\sum_{t=1}^{T} \hat{\rho}_t}$ computes effective sample size where $M$ is the number of chains, $N$ is the number of samples in each chain and $\rho_t$ denotes auto-correlation at lag $t$. For the results reported below $N = 100$ and $M = 1$. Each chain was run with 500 warm-up iterations for the sampler to adapt to an optimal step-size. $\mathtt{ess\_bulk}$ refers to the effective sample size based on the rank normalized draws and is a useful indicator of sampling efficiency. $\mathtt{ess\_tail}$ computes the minimum of the effective sample sizes of the 3% and 97% quantiles [Vehtari et al., 2021].

## B.3   Boston

| hyper | mean | sd | hdi_3% | hdi_97% | mcse_mean | mcse_sd | ess_bulk | ess_tail |
|-------|------|-----|--------|---------|-----------|---------|----------|----------|
| ls[0] | 2.415 | 0.627 | 1.464 | 3.685 | 0.073 | 0.055 | 97.0 | 58.0 |
| ls[1] | 7.236 | 1.641 | 4.814 | 10.737 | 0.137 | 0.105 | 165.0 | 71.0 |
| ls[2] | 5.751 | 1.56 | 2.903 | 8.099 | 0.169 | 0.13 | 104.0 | 69.0 |
| ls[3] | 8.42 | 1.71 | 6.041 | 11.732 | 0.154 | 0.109 | 120.0 | 112.0 |
| ls[4] | 3.711 | 1.308 | 1.708 | 6.042 | 0.141 | 0.1 | 89.0 | 113.0 |
| ls[5] | 3.366 | 0.402 | 2.731 | 4.155 | 0.035 | 0.025 | 141.0 | 78.0 |
| ls[6] | 5.594 | 1.235 | 3.363 | 7.957 | 0.117 | 0.086 | 109.0 | 60.0 |
| ls[7] | 3.078 | 0.926 | 1.524 | 4.89 | 0.092 | 0.065 | 97.0 | 77.0 |
| ls[8] | 6.53 | 1.515 | 4.034 | 9.074 | 0.148 | 0.106 | 104.0 | 62.0 |
| ls[9] | 2.416 | 0.64 | 1.425 | 3.8 | 0.054 | 0.039 | 146.0 | 78.0 |
| ls[10] | 5.388 | 1.505 | 2.815 | 8.137 | 0.143 | 0.106 | 108.0 | 74.0 |
| ls[11] | 6.239 | 2.198 | 3.307 | 10.616 | 0.267 | 0.203 | 75.0 | 77.0 |
| ls[12] | 1.808 | 0.316 | 1.248 | 2.34 | 0.036 | 0.026 | 85.0 | 77.0 |
| sig_f | 1.067 | 0.15 | 0.796 | 1.333 | 0.017 | 0.013 | 75.0 | 59.0 |
| sig_n | 0.277 | 0.01 | 0.261 | 0.293 | 0.001 | 0.001 | 180.0 | 87.0 |

### B.4 Yacht

| hyper | mean | sd | hdi_3% | hdi_97% | mcse_mean | mcse_sd | ess_bulk | ess_tail |
|-------|------|------|--------|---------|-----------|---------|----------|----------|
| ls[0] | 7.486 | 1.042 | 5.522 | 9.25 | 0.049 | 0.037 | 500.0 | 343.0 |
| ls[1] | 10.361 | 1.496 | 7.657 | 13.045 | 0.067 | 0.048 | 498.0 | 320.0 |
| ls[2] | 15.365 | 2.588 | 10.592 | 19.864 | 0.103 | 0.075 | 643.0 | 397.0 |
| ls[3] | 12.464 | 2.12 | 8.818 | 16.494 | 0.073 | 0.053 | 836.0 | 499.0 |
| ls[4] | 15.372 | 2.543 | 10.137 | 19.902 | 0.092 | 0.066 | 740.0 | 307.0 |
| ls[5] | 1.368 | 0.088 | 1.215 | 1.536 | 0.005 | 0.003 | 360.0 | 427.0 |
| sig_f | 2.334 | 0.341 | 1.765 | 3.051 | 0.019 | 0.014 | 323.0 | 382.0 |
| sig_n | 0.034 | 0.002 | 0.03 | 0.038 | 0.0 | 0.0 | 562.0 | 423.0 |

### B.5 Concrete

| hyper | mean | sd | hdi_3% | hdi_97% | mcse_mean | mcse_sd | ess_bulk | ess_tail |
|-------|------|------|--------|---------|-----------|---------|----------|----------|
| ls[0] | 3.667 | 0.537 | 2.86 | 4.776 | 0.059 | 0.042 | 80.0 | 77.0 |
| ls[1] | 5.278 | 0.741 | 4.125 | 6.865 | 0.093 | 0.066 | 65.0 | 117.0 |
| ls[2] | 5.558 | 1.264 | 3.415 | 7.863 | 0.135 | 0.095 | 100.0 | 64.0 |
| ls[3] | 2.933 | 0.497 | 2.19 | 3.976 | 0.054 | 0.041 | 104.0 | 98.0 |
| ls[4] | 3.757 | 0.636 | 2.897 | 4.969 | 0.069 | 0.049 | 81.0 | 77.0 |
| ls[5] | 8.633 | 1.716 | 5.898 | 11.525 | 0.148 | 0.112 | 142.0 | 38.0 |
| ls[6] | 4.453 | 0.624 | 3.324 | 5.633 | 0.065 | 0.047 | 95.0 | 78.0 |
| ls[7] | 1.037 | 0.085 | 0.877 | 1.2 | 0.012 | 0.008 | 51.0 | 78.0 |
| sig_f | 1.588 | 0.242 | 1.187 | 1.977 | 0.032 | 0.023 | 58.0 | 96.0 |
| sig_n | 0.307 | 0.009 | 0.293 | 0.323 | 0.001 | 0.001 | 99.0 | 52.0 |

### B.6 Energy

| hyper | mean | sd | hdi_3% | hdi_97% | mcse_mean | mcse_sd | ess_bulk | ess_tail |
|-------|------|------|--------|---------|-----------|---------|----------|----------|
| ls[0] | 2.788 | 1.231 | 1.215 | 5.005 | 0.114 | 0.081 | 93.0 | 117.0 |
| ls[1] | 3.738 | 1.886 | 1.459 | 7.848 | 0.233 | 0.172 | 89.0 | 102.0 |
| ls[2] | 0.887 | 0.073 | 0.763 | 1.04 | 0.006 | 0.005 | 128.0 | 44.0 |
| ls[3] | 2.92 | 1.186 | 1.209 | 5.078 | 0.131 | 0.093 | 73.0 | 78.0 |
| ls[4] | 2.892 | 1.426 | 1.056 | 5.868 | 0.153 | 0.108 | 100.0 | 67.0 |
| ls[5] | 25.615 | 3.875 | 19.263 | 32.822 | 0.367 | 0.26 | 99.0 | 75.0 |
| ls[6] | 1.93 | 0.165 | 1.668 | 2.261 | 0.021 | 0.015 | 65.0 | 78.0 |
| ls[7] | 21.52 | 2.68 | 16.378 | 26.616 | 0.228 | 0.164 | 139.0 | 78.0 |
| sig_f | 1.002 | 0.134 | 0.795 | 1.206 | 0.016 | 0.011 | 74.0 | 76.0 |
| sig_n | 0.045 | 0.001 | 0.043 | 0.048 | 0.0 | 0.0 | 86.0 | 93.0 |

## B.7 WineRed

| hyper | mean | sd | hdi_3% | hdi_97% | mcse_mean | mcse_sd | ess_bulk | ess_tail |
|-------|------|------|--------|---------|-----------|---------|----------|----------|
| ls[0] | 2.867 | 0.803 | 1.637 | 4.057 | 0.085 | 0.064 | 135.0 | 77.0 |
| ls[1] | 4.048 | 0.947 | 2.594 | 5.948 | 0.067 | 0.056 | 163.0 | 91.0 |
| ls[2] | 4.101 | 1.291 | 2.474 | 6.917 | 0.104 | 0.078 | 170.0 | 77.0 |
| ls[3] | 6.32 | 2.018 | 3.007 | 9.72 | 0.239 | 0.17 | 200.0 | 52.0 |
| ls[4] | 3.806 | 1.135 | 1.815 | 5.552 | 0.096 | 0.073 | 143.0 | 77.0 |
| ls[5] | 6.096 | 2.036 | 3.288 | 10.548 | 0.18 | 0.149 | 196.0 | 59.0 |
| ls[6] | 3.99 | 1.029 | 2.415 | 6.372 | 0.137 | 0.097 | 67.0 | 65.0 |
| ls[7] | 5.925 | 1.667 | 3.177 | 8.986 | 0.189 | 0.139 | 74.0 | 102.0 |
| ls[8] | 4.065 | 1.405 | 1.894 | 6.46 | 0.166 | 0.132 | 109.0 | 55.0 |
| ls[9] | 1.929 | 0.351 | 1.333 | 2.573 | 0.038 | 0.028 | 104.0 | 52.0 |
| ls[10] | 2.58 | 0.453 | 1.765 | 3.471 | 0.045 | 0.033 | 118.0 | 77.0 |
| sig_f | 0.698 | 0.095 | 0.547 | 0.875 | 0.013 | 0.01 | 76.0 | 34.0 |
| sig_n | 0.749 | 0.017 | 0.716 | 0.777 | 0.001 | 0.001 | 188.0 | 102.0 |