# OpenReview forum: "Sparse Gaussian Process Hyperparameters: Optimize or Integrate?"
_NeurIPS.cc/2022/Conference — NeurIPS 2022 Accept_

### Official Review · Reviewer_TWYS · 2022-06-30

**Rating:** 7
**Confidence:** 2
**Soundness:** 4 excellent
**Presentation:** 3 good
**Contribution:** 3 good

**Summary:**

The paper addresses the problem of hyperparameter optimization for sparse GP kernels. The proposed solution utilises MCMC to keep the hyperparameter posterior, accounting for uncertainty in the hyperparameter space. The paper introduces a series of decisions, which improve computational efficiency (separating sampling of Zs and \thetas) and help with handling high-dimensional inputs (by using a doubly collapsed ELBO). The experiments show improvement over alternative techniques.

**Questions:**

Minor issues:
Equation 2 is a little bit weird in a sense that \theta is missing after the first equal sign. I think you should include the dependency on \theta there for p(y|f) and p(f) too.

**Limitations:**

Yes.

**Strengths And Weaknesses:**

Strengths: The proposed fully Bayesian sparse GP is suitable for larger datasets and has more efficient sampling scheme that the alternatives. The proposed approach is theoretically sound.

The problem is well motivated and presented. The authors nicely demonstrate the problem of kernel hyperparameter optimization with Figure 1 and further compare relevant works and alternative approaches with Tables 1 and 2.

Weaknesses: It would be interesting to see more comparison methods in the experiments section. The benchmark datasets could be better motivated, e.g. by explaining their main differences, adding their descriptions.

---

> ### Author Response · Authors · 2022-08-02
> **Response to Reviewer Feedback**
>
> Thank you for the positive feedback and comments.
>
> > Weaknesses: It would be interesting to see more comparison methods in the experiments section. The benchmark datasets could be better motivated, e.g. by explaining their main differences, adding their descriptions.
>
> We add one additional method (across all datasets and 10 splits) inspired by Rossi et al (2021) which treat the inducing inputs $Z$ in addition to $\theta$ in a Bayesian way using SGHMC. We show these results in the general comment above and on page 10 in the revised manuscript.
>
>
> > Minor issues: Equation 2 is a little bit weird in a sense that $\theta$ is missing after the first equal sign. I think you should include the dependency on $\theta$ there for $p(y|f)$ and $p(f)$ too.
>
> We agree the dependency on $\theta$ should be explicit in the prior. We have changed
> to $p(f|\theta)$. The data likelihood $p(y|f)$ only depends on the noise variance $\sigma^{2}$ but since $\sigma^{2} \in \theta$ we are happy to change it to $p(y|f,\theta)$.

---

> > ### Comment · Reviewer_TWYS · 2022-08-09
> > **Final assesment**
> >
> > Thank you for the comments, I believe that the additional method comparison strengthens the paper. My assessment of the paper remains positive.

---

> > > ### Author Response · Authors · 2022-08-09
> > > **Response to Rebuttal feedback (TWYS)**
> > >
> > > Thank you for the positive feedback and response.
> > > The additional experiments suggested have certainly helped improve the quality of the paper. One of the main motivations for the benchmarks used is the ease of comparison and cross-verification with other works in GP literature. The Co2 case study would be fully described and motivated in the manuscript beyond the summary of results here.

---

### Official Review · Reviewer_uTXg · 2022-07-10

**Rating:** 6
**Confidence:** 3
**Soundness:** 4 excellent
**Presentation:** 2 fair
**Contribution:** 3 good

**Summary:**

This paper proposes a new extension of the variational sparse GP framework (Hensman et al., 2015b) to bypass the need of sampling inducing points. This is achieved by a clever analysis of the optimal variational distribution of hyper-parameter $\theta$, which then reveals a way to collapse the inducing points $u$ from the ELBO. This doubly collapsed ELBO is intractable, but its gradient can be estimated via sampling $\theta$ from the optimal variational distribution $q^{\ast}(\theta)$ using HMC. This is a significant contribution for inference with high-dimensional data, since it improves the sampling efficiency. The proposed method achieves promising empirical results on several benchmarks.

**Questions:**

- The warm start strategy was introduced in algorithm 1, but was never described anywhere else. Can the authors elaborate?
- Why do SGPR + HMC and SVGP use different sampler implementations? Could it be that the runtime/RMSE differences are partly due to this choice?
- The latest benchmarks included in the empirical study were SVGP (Hensman et al., 2015a) and JointHMC (Hensman et al., 2015b). Since SGPR+HMC extends these methods, I believe it is appropriate to include them as benchmarks. However, since the introduction of SVGP there have been plenty of other sparse GPs. Table 1 even listed a few, such as Rossi et al., 2021 and Bui et al., 2018. I think this paper should additionally compare to these benchmarks to fully showcase the relevance of the proposed method in light of more recent developments.
- Since I'm not entirely clear about Eq. (14), I'm tentatively putting a score of 5. I will upgrade my score once the authors address my confusion.

**Limitations:**

This is a purely theoretical work. I do not foresee any potential negative societal impacts.

**Strengths And Weaknesses:**

The paper is clearly written on a high level, but some details are not quite clear to me. For instance, I do not quite follow what happens at Eq. (14), which was written with extreme brevity. Where does the addition come from? If I understand correctly, it should be an equality there as we substitute $\mathcal{L}^{\ast\ast}$ with $\mathcal{L}^{\ast}$ evaluated at $q=q^{\ast}$. Then, we apply chain rule to get the inner product. Still, I'm quite confused by the final approximation. How do we get to $\frac{1}{J}\sum_{j=1}^{J} \mathcal{L}_{\Phi,\theta_j}$ ? The rest of the math seems to check out, so maybe the authors can help to clarify this part.
Minor issue: In line 120, it should be plugging back to the Eq. above Eq. (5), which is unlabeled.

As far as I am aware, this work is novel. I really like the clever idea of doubly collapsing the ELBO to avoid sampling inducing inputs. I can see that it would be a problem for inference with high-dimensional data, as we have to draw samples from a very large state space. This was explicitly pointed out by the authors: "While this approach is quite general, in the case of Gaussian regression, it vastly increases the dimensionality of the state space over which HMC must be run relative to HMC in GPR since the u are sampled in addition to the θ". However, I find it quite disappointing that the authors only demonstrate on fairly low-dimensional datasets (e.g., the largest was 13-dimensional). I do not see how these benchmarks justify the use of this method.

----------
I have read the author's rebuttal, which clarified some of my concerns. My overall score will be adjusted to 6. However, I would encourage the authors to continue to revise the empirical study by expanding the range of benchmarks and adding more ablation study to confirm the expected advantage over other methods.

---

> ### Author Response · Authors · 2022-08-02
> **Response to reviewer (1/2)**
>
> Thank you for the feedback and questions, we try and address the comments below:
>
> #### Clarification regarding eq. (14)
>
> > Where does the addition come from? If I understand correctly, it should be an equality there as we substitute $\mathcal{L}^{\ast}$ with $\mathcal{L}^{\ast}$ evaluated at $q = q^{\ast}$
>
> $\mathcal{L}^{\ast\ast}_{\phi} = \mathcal{L}^{\ast}_\phi(q^\ast(\theta))$, which depends on $\phi$ in two ways: $\mathcal{L}^*_\phi$ depends directly on $\phi$, and $q^{\ast}(\theta)$ depends on $\phi$ because it maximises $\mathcal{L}^{\ast}_\phi$.
>
> >Still, I'm quite confused by the final approximation. How do we get to $\frac1J \sum_{j=1}^J
>         \frac{\partial}{\partial\phi} %\left(
>             \mathcal{L}({\phi, \theta_j})$
>
> The result $\frac1J \sum_{j=1}^J
>         \frac{\partial}{\partial\phi} %\left(
>             \mathcal{L}({\phi, \theta_j})$ is obtained by doing a Monte Carlo approximation of $\int q(\theta) \mathcal{L}_{\theta,\phi} d\theta$ in (LHS, first-term of eq. 12); the KL disappears because it does not depend on $\phi$.
>
> (The notation $\mathcal{L}^{\ast}$ and $\mathcal{L}^{\ast\ast}$ indicate levels of collapse, so $\mathcal{L}^{\ast}$ incorporates the optimal distribution $q^{\ast}(u|\theta)$ but is uncollapsed wrt $q^{\ast}(\theta)$, $\mathcal{L}^{\ast\ast}$ incorporates the optimal variational distributions for both $q^{\ast}(\theta)$ and $q^{\ast}(u|\theta)$. Also, $\mathcal{L}\_{\phi,\theta}$ is the canonical Titsias ELBO (eq. (6) **without** the variational treatment of hypers $\theta$). The derivative of $\mathcal{L}^{\ast\ast}$ wrt variational params ($\phi$) ends up reducing to $\int q(\theta) \mathcal{L}\_{\theta,\phi} d\theta$, where we have $\mathcal{L}\_{\theta, \phi}$ in closed-form and we can sample from $q^{\ast}(\theta)$ (section 4.1.1 in the revision).)
>
> Note: We use $\phi$ to denote generic variational parameters, in our set-up the only variational parameters are the inducing locations $Z$.
>
>
> > I really like the clever idea of doubly collapsing the ELBO to avoid sampling inducing inputs. I can see that it would be a problem for inference with high-dimensional data, as we have to draw samples from a very large state space. This was explicitly pointed out by the authors: "While this approach is quite general, in the case of Gaussian regression, it vastly increases the dimensionality of the state space over which HMC must be run relative to HMC in GPR since the u are sampled in addition to the θ". However, I find it quite disappointing that the authors only demonstrate on fairly low-dimensional datasets (e.g., the largest was 13-dimensional). I do not see how these benchmarks justify the use of this method.
>
> To clarify, the doubly collapsed algorithm we propose solely samples the hyperparameters $(\theta)$ and optimizes the inducing locations $(Z)$, hence, it is the dimensionality of the hyperparameter space which matters. Indeed the dimensionality of the hyperparameter space is tied to the input dimensionality if we use a lengthscale per dimension as we do with the SE-ARD kernel. For instance, if $X \in \mathbb{R}^{4}$ and we used $M=100$ inducing points, the jointHMC scheme must sample from a 106 (= 100 inducing points + 4 lengthscales + 2 (signal variance and noise variance)) dimensional space  whereas the sampler in our algorithm would only need to sample from a 6-dimensional hyperparameter space. The elevator dataset uses $d=18$ dimensions yielding a 20-dimensional sampling state-space.
>
> >The warm start strategy was introduced in algorithm 1, but was never described anywhere else. Can the authors elaborate?
>
> The NUTS sampler has to be initialised at some values for $Z$ and $\theta$ (they can be very correlated). The warm-start strategy (of optimizing both $(Z, \theta)$ jointly for a few gradient steps) is used to find a good region for the sampler to initialise $\theta$. Note that the sampler itself has a burn-in/tuning phase where other sampling based hyperparameters (like step-size, tree-depth and path length) are tuned. The warm-start is a heuristic which is not strictly necessary but in practice reduces the time it takes for burn-in as the initial values are guided towards regions of higher probability mass and prevents the chains from getting stuck in bad regions. A similar warm-start protocol was also followed in Hensman et al, 2015b.

---

> > ### Author Response · Authors · 2022-08-02
> > **Response to reviewer (2/2)**
> >
> > > Why do SGPR + HMC and SVGP use different sampler implementations? Could it be that the runtime/RMSE differences are partly due to this choice?
> >
> > I think the reviewer means the jointHMC scheme (instead of SVGP) as we don't conduct sampling with SVGP in this work. The jointHMC (Hensman et al, 2015b) scheme uses the gpflow implementation `gpflow.models.SGPMC`[1] (also jointly authors of Hensman et al, 2015b) and it was most straightforward to implement the `SGPR + HMC` scheme in gpytorch with the `pymc3` NUTS sampler. The hardware on which the experiments were conducted was the same and while we agree there might be small differences attributed to the sampler from the different packages the difference in sampling times between the two inference schemes is quite significant and this is mainly attributed to our hybrid scheme of optimising $Z$ combined with sampling $\theta$.
> >
> > [1] Alexander G de G Matthews, Mark Van Der Wilk, Tom Nickson, Keisuke Fujii, Alexis Boukouvalas, Pablo L&#233;on-Villagr&#225;, Zoubin Ghahramani, and James Hensman. Gpflow: A gaussian process library using tensorflow. J. Mach. Learn. Res., 18(40):1–6, 2017.
> >
> >
> > [comment]: <> (perhaps also make some nod to using the authors code to baseline their methods since James and Alex are the creators of gpflow?)
> >
> > > Since the introduction of SVGP there have been plenty of other sparse GPs. Table 1 even listed a few, such as Rossi et al., 2021 and Bui et al., 2018. I think this paper should additionally compare to these benchmarks to fully showcase the relevance of the proposed method in light of more recent developments.
> >
> > We present additional results (in the general comment above) which implements the key idea in Rossi et al., 2021. of sampling $(Z, \theta)$ jointly. Further, Bui et al. (2018) derive a lower bound where $\theta$ is handled variationally $(q(\theta))$ and $q(u)$ remains uncollapsed permitting batch inference.  Overall, they consider the following 3 schemes in a classification setting with a non-Gaussian likelihood:
> >
> > 1) MCMC for both, the latent function $f$ and hyperparameters $\theta$ without any sparsification. (sampling $f$ only arises in the case of non-Gaussian likelihoods while we side-step sampling $f$ by only considering Gaussian likelihoods in which case GPR + HMC (which we compute) would be the closest method)
> >
> > 2) Sparse VI for the latent function $f$ and ML-II for $\theta$ (this would reduce to SVGP + ML-II or SGPR + ML-II for the Gaussian likelihood case).
> >
> > 3) Sparse VI for both $f$ and $\theta$ with inducing points  - we don't explore this as the focus of our work is on sampling $\theta$ within the doubly collapsed formulation.
> >
> > Importantly, all the methods considered use the uncollapsed bound and the focus is also on the streaming setting where the training data arrive sequentially. We are happy to provide a detailed discussion in the manuscript.

---

> > > ### Comment · Reviewer_uTXg · 2022-08-07
> > > **Response to author's rebuttal**
> > >
> > > Dear authors,
> > >
> > > Thank you for the detailed response. I did a re-read of the papers with your clarification and it now makes sense to me. As promised, I have raised my overall evaluation to 6. However, i still feel that the experiment is quite limited in scope even with new results in the rebuttal. While I remain positive of the idea, I do hope to see a more thorough empirical study in the final revision of this paper.
> > >
> > > Also, it would be nice to have some extra discussion regarding the high level differences between this method & other methods that also applied the Bayesian treatment on the inducing inputs. For instance, can the author clarify what is the main practical advantage that the proposed method has over Rossi et al., 2021? I believe this answer should also be incorporated into Sec 3.2 to make the positioning more transparent.

---

> > > > ### Author Response · Authors · 2022-08-09
> > > > **Response to Rebuttal feedback (uTXg)**
> > > >
> > > > Thank you for your detailed response and feedback.
> > > >
> > > > In conjunction with the experiment shown above we will include a more thorough review of the Rossi et al. method contrasting  with our doubly collapsed method. Below we provide a brief summary addressing your question about the practical advantage over Rossi.
> > > >
> > > > The method by Rossi proposes to sample the inducing points $Z, z_{i} \in \mathbb{R}^{d}$ in addition to $(u, \theta)$. Overall, the method samples $(Z, \theta, u)$. In comparison, by not sampling $Z$ and doubly collapsing, our proposed method samples just $\theta$, which is a much lower-dimensional space. For example, if $M = 100$, $d=1$ and there are 5 hyperparameters, then the method by Rossi would sample a 205-dimensional vector, whereas our method would sample just a 5-dimensional vector.
> > > >
> > > > The additional method incorporated in the revision (FBGP) slightly differs from the method by Rossi. Namely, FBGP combines the method by Rossi with our method: FBGP doubly collapses the lower bound _and_ samples $Z$. This means that FBGP samples $(Z, \theta)$, a 105-dimensional vector. Compared to the 5-dimensional $\theta$, this is still a vast increase in dimensionality.
> > > >
> > > > We see from the evaluation of the datasets above that FBGP increases the runtime (training including sampling in the $(Z, \theta)$ space) by a factor of ~3 for most datasets without providing a concomitant improvement in test RMSE or log predictive density.
> > > >
> > > > The Rossi method may be a more useful technique for non-Gaussian likelihoods where the optimal variational distribution $q^{\ast}(u)$ is intractable, but the focus of our work is on the conjugate regression case. Further, Rossi rely on stochastic gradient HMC. There are two effects which come into play here: mini-batching may help reduce the runtime but the dimensionality of the parameter space over which the sampler must operate is now much higher $(Z, u, \theta)$.

---

### Official Review · Reviewer_vWTc · 2022-07-10

**Rating:** 6
**Confidence:** 3
**Soundness:** 3 good
**Presentation:** 3 good
**Contribution:** 3 good

**Summary:**

The paper proposes a novel framework for tackling Gaussian Process (GP) inference by combining sparse GP (based on inducing variables) and Bayesian hyperparameter inference. The proposed method offers great potential to avoid overfitting compared to previous methods that rely on maximizing the marginal likelihood. Experiments include both synthetic data (one-dimensional) and five small/medium-sized UCI benchmarks. Empirical results show that the proposed methods improve existing sparse GP models.

**Questions:**

N/A

**Limitations:**

Yes, the authors have adequately addressed the limitations and potential negative societal impact of their work

**Strengths And Weaknesses:**

Strengths:
- Considering hyperparameter uncertainty in GP inference is a meaningful and valid direction. The paper thoroughly examined existing methods and proposed a practical method to advance this direction, which brings value to GP's applications.
- All contributions are well motivated and related works are well cited and documented. The entire paper is self-contained and the presentation is logically ideal.
- The proposed method is technically correct, with Algorithm 1 helping understand how the model works.
- Empirical results indicate that the proposed model obtains strong performance on a diverse set of tasks, in particular with reasonable running time.

Limitation: In my opinion, a limitation of the method is that the compared methods are relatively old (SGPR in 2009 [1] and SVGP in 2015 [2]). So it would be interesting to see how this work compares to recent GP inference works (or at least discuss the opportunities of adopting the proposed Bayesian hyperparameter inference to recent works).

Reference
1. Titsias, Michalis. "Variational learning of inducing variables in sparse Gaussian processes." Artificial intelligence and statistics. PMLR, 2009.
2. Hensman, James, Alexander Matthews, and Zoubin Ghahramani. "Scalable variational Gaussian process classification." Artificial Intelligence and Statistics. PMLR, 2015.

---

> ### Author Response · Authors · 2022-08-02
> **Response to Reviewer**
>
> Thank you for the feedback and comments regarding the paper.
>
> > Limitation: In my opinion, a limitation of the method is that the compared methods are relatively old (SGPR in 2009 [1] and SVGP in 2015 [2]). So it would be interesting to see how this work compares to recent GP inference works (or at least discuss the opportunities of adopting the proposed Bayesian hyperparameter inference to recent works).
>
> Regarding the comparisons to more recent methods, while SGPR and SVGP are both reasonably old methods, they are still competitive on many tasks with more modern methods. Minor variations on SGPR (e.g. the collapsed version of SOLVE-GP [1] or CGLB [2]) are roughly state-of-the-art on many tasks for scalable model selection in many instances, and generally perform comparably to SGPR. Additionally, the core idea of collapsing the bound we emphasize in this work would generalize to SOLVE-GP in a straight-forward manner as it has an interpretation as variational inference (just using (eqn 15, Appendix A [1] in place of eqn 6). Similarly, one could use the CGLB variational bound in an MCMC scheme to estimate the likelihood, although the method would not have a clear variational interpretation and so one might be concerned with selecting inducing points in this manner. Both of these variations would lead to an at least quadratic cost in the dataset size, which may be an obstacle for particularly large datasets.
>
> An alternative comparison would be to conjugate gradient based methods. While perhaps relevant, these are difficult to combine with HMC schemes as they only give stochastic estimate of the log marginal likelihood and its gradient (these arise due to the use of stochastic trace estimators and related ideas). We can add a comparison to type-II maximum likelihood selection of $\theta$ using conjugate gradient based approaches implemented in GPyTorch.
>
> [1] Jiaxin Shi, Michalis K. Titsias, and Andriy Mnih. Sparse Orthogonal Variational Inference for Gaussian Processes. AISTATS 2020. https://arxiv.org/abs/1910.10596
>
> [2] Artem Artemev, David R. Burt, Mark van der Wilk. Tighter Bounds on the Log Marginal Likelihood of Gaussian Process Regression Using Conjugate Gradients. ICML 2021. https://proceedings.mlr.press/v139/artemev21a.html

---

### Official Review · Reviewer_gGgw · 2022-07-11

**Rating:** 6
**Confidence:** 3
**Soundness:** 3 good
**Presentation:** 2 fair
**Contribution:** 3 good

**Summary:**

The paper contributes a specific inference method for sparse variational Gaussian process regression in a Bayesian setup allowing inference of  hyperparameters which doesn't require sampling the inducing variables u. The new algorithm is derived by exploiting the exact bound derived in Titsias, 2009 to "collapse" the full ELBO (with hyperarameters) leading to a more effective algorithm in the Bayesain setting.

The algorithm is evaluated on a relevant set of synthetic and benchmark regression problems which demonstrates that the proposed method outperforms existing sparse inducing points methods in terms of RMSE, predictive log-likelihood, and wall clock time. It is not clear that the approach consistently outperforms a basic SGPR+MLII approach.

**Questions:**


Comments/suggestions/questions:
- l105 mentions the classification case, although only regression with the Normal likelihood is considered going forward. I'd suggest removing or clarifying.

- Table 2: I am not sure I follow what's going on in the table; e.g. how is the "quality" determined? Why does MAP not support a non-Gaussian likelihood?... I'd suggest adding a reference per entry where these facts are extracted unless clearly explained e.g. in the appendix. Also, perhaps explicitly indicate which approach is yours.

- Algorithm 1 (page 6): I am slightly confused by the comments and notes in the algorithm box (line 2-3 and 11); are the kernel hyperparameters part of the optimization or not? Additionally, I think it would be beneficial to include reference to the relevant equations (specifically for the \mathcak{L} elements )

- Figure 2: Please clarify the caption; I (initially) found it difficult to work out exactly what was happening in the various panels.

- Figure 3: There are several SGPR+HMC variants; I assume the authors prefer the reader to focus on "Adapt 2" but it would be helpful with some guidance.

- Table 4: $\delta$ is not explained/defined

- Sec 5.2: I believe the discussion fails to comment on the performance of SGPR (ML-II) vs. the proposed SGPR+HMC (Adapt 2) as observed in table 4 and figure 4? Would this not suggest that the benefit of a fully Bayesian approach is not evident in most cases?

- l250 (figure on the right): It would be informative with the GPR+HMC and SGPR+HMC performance shown as horizontal lines.

Very minor:
- Conditioning on X seems a bit arbitrary and briefly introduced in 101-108, then disappears in sec 3.1, but back in l130... I'd suggest more consistency/clarity.

- l101 vs introduction: the text switches from "kernel" to "covariance function" and it is used interchangeably throughout; perhaps consider if a consistent use would be beneficial

- Eq (1) missing comma before f

- l151: "...sample the u..." -> "...sample u...."

- l190: check missing punctuation prior to  "...the outer..."

- l234: I am not sure where this common practice is used; could you provide a reference?

- l248, l266: "don't" -> "do not" I would suggest avoiding contractions.

- l261: not sure what "this" refers to





**Limitations:**

See points made elsewhere regarding the discussion of results etc.



------------------------
Following the authors' response, I have updated my score to 6.

**Strengths And Weaknesses:**



- Novelty and signiificance: The approach appears to be leverage a new insights about the ELBO and contributes a sensible addition to the sparse GP regression toolbox. The paper provides a better understanding of GP inference in the sparse regime which is valuable in its own right, although it is arguably not very far reaching and significant from a practical point of view in part because many people simply resort to MLII which also seems to perform on par with the proposed method on most of the UCI benchmarks.
- Quality: The basic idea appears sensible and the core algorithmic work well-executed. I find that the experiments could have been presented and discussed better (see specific suggestions/questions below).
- Clarity and presentation: I find that there are several questions (see below) that I would like to see addressed in order for it to be easily accessible and comphrensive.

---

> ### Author Response · Authors · 2022-08-02
> **Response to Reviewer gGgw**
>
> Thank you for the detailed feedback and questions. We try to address the questions and comments below:
>
> > l105 mentions the classification case, although only regression with the Normal likelihood is considered going forward. I'd suggest removing or clarifying.
> >
> We agree with the reviewers comment that the background section should focus on the conjugate regression case as that is the focus of the paper.
>
> #### Q: Clarification for Table 2
>
> A: Quality is being used in an asymptotic (in compute) sense, and represents the ability of the method to faithfully represent the posterior distribution over hyperparameters. To elaborate, if VI is run to convergence, a potentially significant amount of error will be incurred by the Gaussian approximation to the non-Gaussian posterior over the hyperparameters (red). On the opposite extreme, if no sparsity assumption is made, MCMC over the hyperparameters without sparse approximations is asymptotically consistent (green). The inducing point approximations combined with MCMC lie somewhere inbetween these methods (yellow). For a fixed amount of computation quality of each method my differ (e.g. VI may be preferable in some instances on a fixed budget to MCMC based methods). We will add detail to the caption to clarify the meaning of this column.
>
> We agree that adding key references to the table is needed and we have done that in the revised version.
>
> > Algorithm 1 (page 6): I am slightly confused by the comments and notes in the algorithm box (line 2-3 and 11); are the kernel hyperparameters part of the optimization or not? Additionally, I think it would be beneficial to include reference to the relevant equations (specifically for the $\mathcal{L}$ elements)
>
> Line 11: We freeze the hyperparameters ($\theta$) at the start of the training loop as we don't want to optimize them. $\theta$ are sampled (using NUTS) from the optimal variational distribution $q^{\ast}(\theta)$ at pre-specified intervals ($t \bmod L == 0$). These samples are used to compute a stochastic estimate of the ELBO $\mathcal{\hat{L}}$ (line 22 in original, 23 in revised).
>
> Apologies for the confusion, we acknowledge the need for clarity in notation here. Lines 2-3 describe the initialisation set-up, we need some starting values for variational parameters ($Z$) and hyperparameters ($\theta$) which can be random but we use the sub-routine warm-start which optimizes the ELBO $L_{\theta, Z}$ w.r.t $(Z, \theta)$ for a few gradient steps prior to the start of the training loop. ($\mathcal{L}_{\theta, Z}$ is eq. 6, the Titsias canonical ELBO, we write this less succintly as $\mathcal{L}(\theta, Z)$ in the algorithm to make the arguments clearer).
>
> Overall, the core training algorithm alternates between two steps:
>
> Sampling step for $\theta$: $\theta_{j} \sim q^{\ast}(\theta)$ and
> Optimisation step for $Z$:  $Z_{opt} \longleftarrow \texttt{optim}(\mathcal{\hat{L}})$,  where  $\mathcal{\hat{L}} = \mathbb{E}\_{q^{\ast}}
> (\theta)[\mathcal{L}(\theta, Z)]$ $\approx \dfrac{1}{J}\sum\_{j=1}^{J}\mathcal{L}(\theta\_{j}, Z\_{opt})$.
>
> Crucially, we can sample from $q^{\ast}(\theta)$ as we can compute it pointwise upto a normalising constant (lines 159-167, eq. 11, 12 and 13). Further, the log of the target distribution (the unnormalized target density for sampling) is just the canonical Titsias ELBO (eq. 6) with the log prior of the hypers $\theta$.
>
> $$\log q^{\ast}(\theta) = \dfrac{1}{C}\log(M_{\theta}p(\theta)) \propto \mathcal{L}({\theta, Z_{opt}}) + \log p(\theta)$$
>
> We have clarified the notation in Algorithm 1 in the manuscript, re-organised the steps and added further comments for better understanding. Note that the clarifications for lines 2-3 and line 11 above refer to the original manuscript not the latest revised one.
>
> > Figure 2: Please clarify the caption; I (initially) found it difficult to work out exactly what was happening in the various panels.
>
> The upper-row shows the posterior predictive distribution under the 3 different inference schemes shown in table 3. We will clarify that the SGPR+HMC scheme is the doubly collapsed scheme described in Algorithm 1. The middle panel (our scheme) in this example yields better generalisation in regions where the input is scarce (reflected in lower test RMSE) and better uncertainty quantification (higher coverage in regions without training data, this is reflected in the lower NLPD.) The bottom row (middle) depicts the histogram of lengthscales drawn from the hyperparameter posterior using our scheme compared to ML-II, given that ML-II converges to a much lower lengthscale with weaker test error and log predictive density indicates it is overfitting. The bottom right plot shows that the ML-II underpredicts the true noise level in the data (a classic sign of overfitting) while the samples from the posterior converge around the ground truth value.

---

> > ### Author Response · Authors · 2022-08-02
> > **(cont., part II) Response to Reviewer gGgW**
> >
> > > Figure 3: There are several SGPR+HMC variants; I assume the authors prefer the reader to focus on "Adapt Z" but it would be helpful with some guidance.
> >
> > We will clarify that the doubly collapsed algorithm described in the manuscript refers to SGPR + HMC (Adapt Z) where the inducing locations are optimized instead of being fixed. The "Fix Z" version fixes the inducing locations to a subset of the training set and does not optimize them during training.
> >
> > > Table 4: $\delta$ is not explained/defined
> >
> > $\delta$ was meant to indicate that the posterior distribution over hyperparameters is approximated by a point estimate (essentially, ML-II inference). We will clarify this in the caption.
> >
> > #### Q: Comparison between SGPR+ML-II and SGPR+HMC.
> >
> > A: On the moderate to smaller datasets shown in Table 4 and Fig. 4 the advantages of ML-II vs. the fully Bayesian scheme are quite subdued in terms of both the RMSE and NLPD. All the datasets use an identical convention of the SE-ARD kernel and M=100 inducing points. We believe that datasets with more sophisticated kernels and fewer inducing points brings out the adavantages of the HMC scheme further. To this end we include an example on the CO$_{2}$ Mauna Loa a well-studied benchmark in GP literature (see Rasmussen and Williams §5.4.3). The time-series demands a highly structured additive kernel with 4 components modeling different aspects of the data. Even though this is a 1d problem, it yields a 13 dimensional hyperparameter space due to the additive kernel which makes for an interesting case-study. We compare our scheme (SGPR + HMC) with ML-II inference. SGPR + HMC gives a favourable trade-off in terms of uncertainty quantification and accuracy compared to running exact GPR HMC and clearly outperforms ML-II.
> >
> > > l250 (figure on the right): It would be informative with the GPR+HMC and SGPR+HMC performance shown as horizontal lines.
> >
> > We are happy to render the results horizontally.
> >
> > Thank you for the other minor comments which we have now fixed in the revision. To address some specifically,
> >
> >
> > > l234: I am not sure where this common practice is used; could you provide a reference?
> >
> > Sure, the UCI benchmarks used in our paper have been used with a 80/20 train/test split in [1,2] and the normalisation and scaling protocol have been used in [3].
> >
> > > l261: not sure what "this" refers to in "*While it is possible to train Exact GPR with HMC for datasets of **this** size...*"
> >
> > We mean it is possible to run exact GP inference with HMC (GPR + HMC) using the marginal likelihood objective instead of sparse variational inference with HMC (SGPR + HMC) as the UCI benchmarks considered are all under 2000 datapoints (N<2000) it is instructive to look at how much longer it takes versus the accuracy lost in using sparsification. This is exactly what Fig.4 encapsulates.
> >
> > References:
> >
> > [1] Simone Rossi, Markus Heinonen, Edwin Bonilla, Zheyang Shen, and Maurizio Filippone. Sparse Gaussian processes revisited: Bayesian approaches to inducing-variable approximations. In *International Conference on Artificial Intelligence and Statistics*, pages 1837–1845. PMLR, 2021
> >
> > [2] Marton Havasi, Jos&#233; Miguel Hern&#225;ndez-Lobato, and Juan Jos&#233; Murillo-Fuentes. Inference in Deep Gaussian Processes using Stochastic Gradient hamiltonian monte carlo. *Advances in Neural Information Processing Systems*, 31, 2018.
> >
> > [3] Hugh Salimbeni and Marc Deisenroth. Doubly Stochastic Variational Inference for Deep Gaussian Processes. *Advances in Neural Information Processing Systems*, 30, 2017

---

> > > ### Comment · Reviewer_gGgw · 2022-08-08
> > > **Response to the authors' rebuttal**
> > >
> > > Dear authors,
> > > Many thanks for your detailed response to my questions and comments.
> > >
> > > I am generally satisfied that the comments relating to clarity (also from other reviewers) have been largely addressed.
> > >
> > > I am happy to see the CO2 example, which at first glance appears to provide more convincing evidence that the suggested approach can also make a practical difference in quality compared to a basic MLII approach. What's the wall clock time for the MLII approach? If possible, would you be able to provide the standard error relating to the CO2 example?
> > >
> > > A few late suggestions: I think the critical wall clock vs. NLPD/RMSE narrative could be better supported better if integrating Fig 3+4. E.g., I could imagine a simple scatter plot (perhaps with some error bars) with wall clock vs. NLPD with the various methods. Either way, I think the MLII wall clock should be included in Figure 4 for completeness, given that it seems to be the closest competitor in terms of NLPD on the UCI benchmarks.
> > >
> > > Overall, I feel more optimistic about the paper after the authors' response, and I will likely raise my score; however, I also agree with other reviewers that more convincing examples (similar to the CO2 example) would likely be needed to ensure that the paper has a sufficient impact.

---

> > > > ### Author Response · Authors · 2022-08-09
> > > > **Response to Rebuttal feedback (gGgW)**
> > > >
> > > > Thank you for the detailed response and feedback.
> > > >
> > > > We agree that the benefits of the proposed method should be contextualised along with the compute cost. Note that we distinguish between two types of compute time. 1) only the time it takes for sampling from the intractable posterior over unknowns approximated by $q^{\ast}(\theta)$ in our method (SGPR + HMC) and $p(\theta, u)$ in JointHMC and 2) the overall training runtime.
> > > >
> > > > Fig. 4 focuses on the sampling time hence the methods included are all sampling based. ML-II and SVGP do not rely on sampling which is why they are excluded from the plot.
> > > >
> > > > However, we are happy to report the training run-times for SGPR (4000 iterations) with ML-II below along with our method (SGPR + HMC), and GPR + HMC, FBGP for context.
> > > >
> > > > | Dataset       |  Boston         |  Concrete        |  Energy           |  WineRed          |  Yacht           |
> > > > |---------------|-----------------|------------------|-------------------|-------------------|------------------|
> > > > | SGPR (ML-II)  |  22.17 (0.21)   |  33.06 (0.07)    |  30.36 (0.09)     |  39.96 (0.87)     |  20.41 (0.22)    |
> > > > | SGPR + HMC    |   29.47 (0.34)  |  53.85 (2.36)    |  61.60 (1.47)     |  60.65 (0.63)     |  24.50 (0.40)    |
> > > > | GPR + HMC     |   78.05 (2.36)  |  977.40 (13.82)  |  326.180 (15.87)  |  1426.25 (39.49)  |  31.71 (0.59)    |
> > > > | FBGP          |  72.63 (8.29)   |  156.31 (4.30)   |  259.81 (11.58)   |  175.45 (13.14)   |  101.92 (2.27)   |
> > > >
> > > > It is important to note that there is a significant amount of flexibility in the experimental design in terms of the number of tuning steps (in the sampler), length of the warm-start, how frequently to interleave the sampling windows etc. All these factors affect the overall running time of the algorithm. We report the run times for the settings described in the appendix.
> > > >
> > > > The $CO_{2}$ dataset is a time-series dataset of emissions in ppm (parts per million) and we are focussing on extrapolation. We train on the past (fixed data points, ~1958-2010) to predict the future(2011-2019), it is not immediately obvious how to replicate the experiment across data splits (to furnish standard errors) keeping the number of inducing points and other parameters fixed.
> > > >
> > > > One way to design the experiment would be to test on a fixed 5 year time period and train on all penultimate years. For instance, Run 1: train_period: 1958-1990, test_period: 1990-1995, Run 2: train_period: 1958-1995, test_period: 1995-2000 and so on. As the length of the training window grows additional inducing points would be required in each run; this wouldn't result in valid standard errors. It is not trivial to set up this experiment in such a short span of time and we will endeavor to furnish confidence intervals for a reasonable experimental framework for the test RMSE and NLPD in this particular experiment in the final manuscript. It takes 123 seconds to train SGPR with ML-II for the current results which train on years 1958-2010 and test on 2011-2019.

---

### Author Response · Authors · 2022-08-02
**Summary of Author Response**

We thank all the reviewers for their time and detailed feedback. We provide a summary of the responses and changes  to the manuscript below:

1) (Reviewer gGgw and uTXg) We have clarified section 4.1, 4.1.1 which derives the doubly collapsed ELBO. Further, we have  added equation numbers in Algorithm 1 and harmonized the notation with the rest of the document.

2) (Reviewer gGgw) We have added key references to the methods referred to in Table 2 on computational complexity.

3) (Reviewer uTXg and vWTc) We add another benchmark (FBGP for *fully Bayesian GP*) inspired by the method in Rossi et al., 2021 which additionally focuses on sampling the inducing inputs $Z$ as opposed to optimisation. The method we present still differs from Rossi on two points:
- We use NUTS to sample from the posterior over $(Z, \theta)$ vs. stochastic gradient HMC based on minibatches of data.
- Our objective incorporates the optimal Gaussian variational distribution $q(u)$ rather than sampling $u$ free-form. We report mean metrics over 10 splits ($\pm$ standard error of mean) with 20% of the data held-out for testing:

Test RMSE ($\downarrow$ is better)

| Dataset  | SGPR + HMC (ours) | FBGP (new)    |
|----------|-------------------|---------------|
| Boston   | 3.286 (0.090)     | 3.845 (0.103) |
| Concrete | 5.402 (0.052)     | 6.085 (0.111) |
| Energy   | 0.469 (0.009)     | 0.490 (0.011) |
| WineRed  | 0.635 (0.008)     | 0.642 (0.007) |
| Yacht    | 0.387 (0.030)     | 0.569 (0.037) |

Test NLPD ($\downarrow$ is better)

| Dataset  | SGPR + HMC (ours) | FBGP (new)    |
|----------|-------------------|---------------|
| Boston   | 2.524 (0.022)     | 2.714 (0.027) |
| Concrete | 3.065 (0.012)     | 3.227 (0.011) |
| Energy   | 0.644 (0.013)     | 0.746 (0.010) |
| WineRed  | 0.961 (0.011)     | 0.973 (0.013) |
| Yacht    | 0.391 (0.13)      | 0.851 (0.061) |

Runtime (Wall clock sec;$\downarrow$ is better)

| Dataset  | SGPR + HMC (ours) | FBGP (new)       |
|----------|-------------------|------------------|
| Boston   | 29.47 (0.34)      | 72.633 (8.296)   |
| Concrete | 53.85 (2.36)      | 156.316 (4.292)  |
| Energy   | 61.60 (1.47)      | 259.816 (11.585) |
| WineRed  | 60.65 (0.63)      | 175.458 (13.141) |
| Yacht    | 24.50 (0.40)      | 101.928 (2.274)  |

We use $M=100$ inducing points for each dataset and use 100 samples for the mixture posterior predictive, we use the same priors over hyperparameters (Appendix A.1) and use a standard Normal prior for the elements of the inducing locations $Z \in \mathbb{R}^{M \times D}$. Our method statistically outperforms FBGP across all datasets. Importantly, sampling $Z$ adds significant overhead to the runtime. It might be the case that $Z$ might benefit from a more heavy tailed prior encouraging more extreme values to be sampled if needed. We don't explore sensitivity to the choice of prior in these experiments.

The code for Rossi is not publicly available. We have contacted that authors to see if we can obtain a copy of the code, and will otherwise implement it as faithfully as possible to compare with for an additional point of comparison.


4) (Reviewer gGgw) We run ML-II, GPR + HMC, SGPR + HMC (our method) on  the CO$_{2}$ dataset to give an example with higher dimensional parameter space, where ML-II can overfit and HMC leads to significant gains.

|                   | GPR + HMC | SGPR + ML-II | SGPR + HMC |
|-------------------|-----------|--------------|------------|
| RMSE              | 1.901     | 2.896        | 2.564      |
| NLPD              | 0.403     | 0.510        | 0.423      |
| Wall clock (sec.) | 2248      | --           | 1003       |

---

### Meta-Review · Area_Chair_HuQ1 · 2022-08-31

**Recommendation:** Accept
**Confidence:** Certain

**Metareview:**

The paper presents a new method for scalable inference in Gaussian regression based on the inducing-variable formalism. It shows that it is possible to sample covariance hyper-parameters while avoiding sampling the inducing variables, a consequence of their doubly collapsed bound. The reviewers agree that it is a technically solid paper and the authors have addressed their concerns satisfactorily, with one of the reviewers raising their scores and the authors providing additional results. I believe this work is worth presenting at NeurIPS and, therefore, recommend its acceptance.

**Award:**

No

---

### Decision · Program_Chairs · 2022-09-14

Accept